EMBO
Molecular Medicine

# Aberrant upregulation of CaSR promotes pathological new bone formation in ankylosing spondylitis

Xiang Li[1,2,†], Siwen Chen[1,2,†], Zaiying Hu[3,†], Dongying Chen[4], Jianru Wang[1,2], Zemin Li[1,2], Zihao Li[1,2], Haowen Cui[1,2], Guo Dai[1,2], Lei Liu[5], Haitao Wang[1,2], Kuibo Zhang[5], Zhaomin Zheng[1,2], Zhongping Zhan[4] & Hui Liu[1,2,*] iD

## Abstract

Pathological new bone formation is a typical pathological feature in ankylosing spondylitis (AS), and the underlying molecular mechanism remains elusive. Previous studies have shown that the calcium-sensing receptor (CaSR) is critical for osteogenic differentiation while also being highly involved in many inflammatory diseases. However, whether it plays a role in pathological new bone formation of AS has not been reported. Here, we report the first piece of evidence that expression of CaSR is aberrantly upregulated in entheseal tissues collected from AS patients and animal models with different hypothetical types of pathogenesis. Systemic inhibition of CaSR reduced the incidence of pathological new bone formation and the severity of the ankylosing phenotype in animal models. Activation of PLCγ signalling by CaSR promoted bone formation both *in vitro* and *in vivo*. In addition, various inflammatory cytokines induced upregulation of CaSR through NF-κB/p65 and JAK/Stat3 pathways in osteoblasts. These novel findings suggest that inflammation-induced aberrant upregulation of CaSR and activation of CaSR-PLCγ signalling in osteoblasts act as mediators of inflammation, affecting pathological new bone formation in AS.

**Keywords** ankylosing spondylitis; CaSR; inflammation; inflammatory cytokines; pathological new bone formation
**Subject Category** Musculoskeletal System

## Introduction

Ankylosing spondylitis (AS) is a chronic inflammatory disease affecting the axial skeleton that is grouped under the term spondylarthritis (SpA) (Taurog *et al*, 2016; Sieper & Poddubnyy, 2017). In addition to inflammatory back pain and arthritic destruction, spinal ankylosis resulting from increasing pathological new bone formation is a typical feature of AS, causing disability and tremendous socioeconomic costs (Taurog *et al*, 2016). Although recent investigations and medications have focused on the aetiology and symptom-controlling therapy, the pathological mechanism of pathological new bone formation is still not clarified. Targeted treatment of axial skeletal ankylosis is demanded because the prognosis so far is unsatisfying (Deodhar, 2018; Molnar *et al*, 2018).

The correlation between inflammation and pathological new bone formation remains enigmatic. Several studies have demonstrated that various types of osteogenic growth factors are involved in pathological new bone formation (Lories & Dougados, 2012; Ruiz-Heiland *et al*, 2012; Gonzalez-Chavez *et al*, 2016). Recently, we demonstrated that inflammation intensity-dependent expression of osteoinductive factors and Wnt family proteins by proinflammatory cells plays a vital role in inflammation-induced pathological new bone formation (Li *et al*, 2018). Thus, consistent with other studies, we confirmed that enhanced differentiation of osteogenic precursor cells induced by growth factors is an important molecular mechanism of inflammation-induced bone formation in the pathological microenvironment. However, other than the indirect effect through induction of osteogenic growth factors, whether inflammatory cytokines have direct positive regulatory effects on osteogenic differentiation of precursor cells remains unknown.

Calcium-sensing receptor (CaSR) is a diametric family of G protein-coupled receptors, which have a critical role in modulating $Ca^{2+}$ homeostasis via its role in the parathyroid glands and kidneys,

1 Department of Spine Surgery, The First Affiliated Hospital, Sun Yat-sen University, Guangzhou, China
2 Guangdong Province Key Laboratory of Orthopaedics and Traumatology, Guangzhou, China
3 Department of Rheumatology and Immunology, The Sixth Affiliated Hospital, Sun Yat-sen University, Guangzhou, China
4 Department of Rheumatology and Immunology, The First Affiliated Hospital, Sun Yat-sen University, Guangzhou, China
5 Department of Spine Surgery, The Fifth Affiliated Hospital, Sun Yat-sen University, Guangzhou, China
*Corresponding author. Tel: +86 138 2609 6992; Fax: +86 20 87331655; E-mail: liuhui58@mail.sysu.edu.cn
†These authors contributed equally to this work

which might affect bone mass. Normally, CaSR modulates systemic $Ca^{2+}$ homeostasis by detecting increasing circulating concentrations of $Ca^{2+}$, leading to intracellular signalling events that mediate reduced parathyroid hormone (PTH) secretion and a reduction in renal tubular $Ca^{2+}$ reabsorption, therefore influencing skeletal homeostasis (Gowen et al, 2000; Hannan et al, 2016).

Moreover, CaSR is abundantly expressed in cells of osteogenic lineage and in skeleton tissues. Several studies have confirmed that CaSR plays a pivotal role in modulating differentiation and mineralization of osteogenic precursor cells (Chang et al, 2008; Hendy & Canaff, 2016; Hannan et al, 2018b). Activation of CaSR downstream pathways in osteoblasts promotes osteogenic differentiation in vitro, while ablating CaSR expression specifically in osteoblasts suppresses osteogenic activity and the mineralizing function in bony calluses and leads to decreased bone mass in animal models (Chang et al, 2008; Dvorak-Ewell et al, 2011; Gonzalez-Vazquez et al, 2014; Cheng et al, 2020).

Interestingly, previous studies showed that hyperstimulation of CaSR is a driving factor of the pathological progression in multiple inflammatory conditions, such as asthma and burn injury (Wu et al, 2015; Yarova et al, 2015; Cheng, 2016; Lee et al, 2017). Some studies reported that several inflammatory cytokines, including IL-6 and IL-1β, drive overexpression of CaSR and worsen the pathological condition in burn injury and sepsis (Hendy & Canaff, 2016; Klein et al, 2016). Given that activation of CaSR and downstream cellular signalling enhances osteogenic differentiation of osteoblasts, while aberrant expression of CaSR is commonly observed under inflammatory conditions, we speculated that CaSR might be involved in the pathological process of pathological new bone formation in the inflammatory microenvironment of AS.

In the current study, we explored the role of CaSR in AS and observed overexpression of CaSR in osteoblasts in spinal entheseal tissues collected from AS patients. Furthermore, aberrant expression of CaSR and activation of its downstream signalling pathways was also confirmed in several classic animal models of AS with different hypothetical types of pathogenesis. Inhibition of CaSR activation attenuated pathological new bone formation phenotype in vivo and osteogenic differentiation in cultured hBMSCs in vitro. Furthermore, various inflammatory cytokines were confirmed to enhance CaSR expression in osteoblasts. Hence, our findings revealed a prominent role for CaSR in the interplay between inflammation and the process of ankylosis progression, which might shed more light on the enigma of inflammation-related pathological new bone formation in AS and propose a potential therapeutic target for slowing ankylosis progression.

# Results

### CaSR+ osteoblasts accumulate in spinal tissues from AS patients

To investigate whether CaSR plays a role in the process of pathological new bone formation in AS, we first assessed expression of CaSR in uncalcified spinal ligament tissues (supraspinous ligament and interspinous ligament) from AS patients who underwent correction surgeries (Fig 1A). Immunohistochemical staining showed that increased CaSR+ cells were accumulated at the entheseal site of the spinal ligament attaching the spinous process from

patients with AS than in age- and sex-matched controls. Cells that expressed Runx2, an osteogenic marker of naive osteoblasts, also accumulated at the same site as CaSR+ cells in the AS group (Fig 1B and C). RT–qPCR and Western blot analysis showed that expression of CaSR was increased in spinal ligament tissues of AS (Fig 1D and E). Immunofluorescence staining revealed that cells highly expressing CaSR were primarily Runx2+ naive osteoblasts (Fig 1F and G). SOFG (Safranine O-Fast Green) staining showed that spinous process and calcified ligament were indistinguishable. Meanwhile, increased cells accumulated at the uncalcified zone in the AS group (Fig 1H). In addition, immunofluorescence staining of the same region showed that the majority of CaSR+ cells expressed OCN, a mature osteoblast marker (Fig 1H and I). These results suggested that CaSR+ osteoblasts accumulated at potential pathological new bone-forming site of enthesis of the spinal ligament in AS patients.

### CaSR+ osteoblasts accumulate at pathological new bone-forming sites in animal models of AS

A proteoglycan-induced spondylitis (PGIS) mouse model that exhibits both axial and peripheral inflammation was established to observe pathologic changes in spinal ankylosis (Hanyecz et al, 2004; Szabo et al, 2005; Adarichev & Glant, 2006; Berlo et al, 2006). The result of μCT analysis showed that spinal ankylosis gradually developed 24 weeks after PG induction (Fig 2A). The incidence of spinal ankylosis per cage increased from 0% to 33.3 ± 11.55% 8 weeks later and continued rising to 73.33 ± 11.55% at 24 weeks (Fig 2B). The bone volume (BV) of pathological new bone was increased 16 and 24 weeks after PG induction (Fig 2C). Meanwhile, RT–qPCR analysis showed that expression of CaSR was increased in spinal ligament tissues of the PGIS model (Fig 2D). H&E staining revealed pathological new bone formation at spinal entheseal sites attaching to the vertebral growth plate at 24 weeks. SOFG staining showed pathological process of new bone formation at 16 and 24 weeks through endochondral ossification (Fig 2E). Immunofluorescence staining revealed that cells expressing CaSR were primarily Runx2+ osteoblasts, which are analogous to human ligament tissues from AS patients. The number of CaSR+ Runx2+ osteoblasts was increased at spinal ligament tissues 16 and 24 weeks after PG induction compared to baseline (Fig 2E and F). Thirty weeks after PG immunization, most CaSR+ cells accumulated at spinal entheseal sites expressed OCN compared to baseline and 8 weeks (Fig 2G and H).

An ageing DBA/1 arthritis animal model spontaneously developed pathological new bone in the hind paws without spinal involvement compared to the PGIS animal model. This model shares multiple similarities with AS in humans, including enthesitis and entheseal new bone formation (Matthys et al, 2003). μCT analysis showed that pathological new bone formation (red area) in the hind paws was increased at 16 and 20 weeks of age (Fig 3A). The incidence of pathological new bone formation in the hind paws per cage increased from 0% at 8 weeks of age to 29.63 ± 6.41% and 74.08 ± 6.41% at 16 and 20 weeks of age, respectively (Fig 3B). BV of pathological new bone was gradually increased at 16 and 20 weeks of age (Fig 3C). RT–qPCR analysis showed that expression of CaSR was increased in hind paw tissues (Fig 3D). H&E and SOFG staining demonstrated pathological new bone formation at

enthesal sites on dorsal surfaces at 16 and 20 weeks of age (Fig 3E). Immunofluorescence staining revealed that cells highly expressing CaSR were mostly Runx2[+] osteoblasts analogous to the phenomenon observed in human ligament tissues from AS patients. The number of CaSR[+] Runx2[+] osteoblasts increased at entheseal sites on dorsal surfaces at 16 and 20 weeks of age (Fig 3E and F).

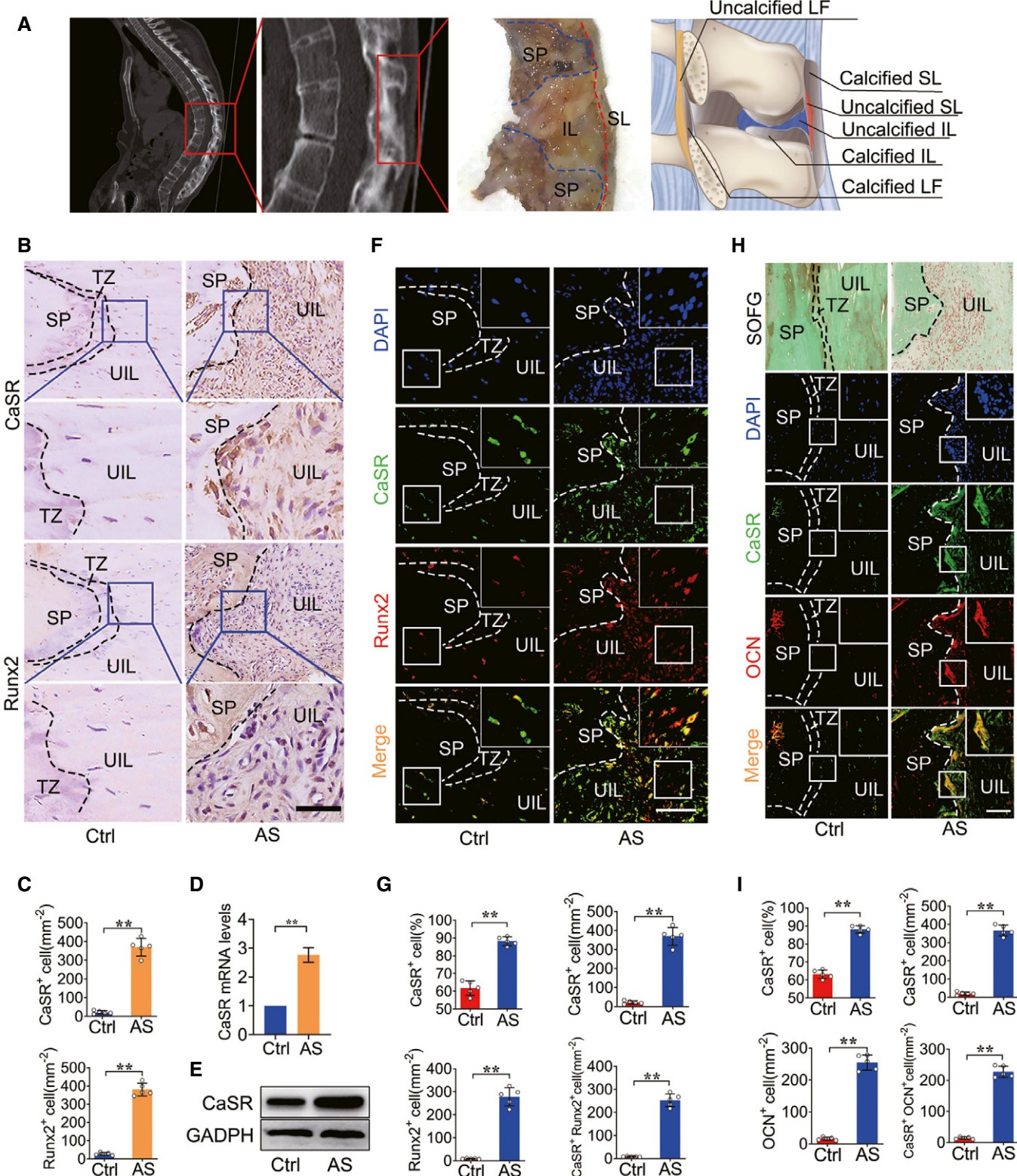

Figure 1.

**Figure 1. CaSR$^+$ osteoblasts accumulated in the spinal entheseal tissues from AS patients.**

A   An illustration of spinal ligament tissues collection.
B   SOFG staining of spinal entheseal tissue and immunohistochemical analysis of CaSR$^+$ and Runx2$^+$ cells in human spinal tissue. The bottom panels show higher magnification of the boxed area in the top panels.
C   Quantitative analysis of the number of CaSR$^+$ and Runx2$^+$ cells per area (mm$^2$). $n = 5$, Student's $t$-test.
D   RT–qPCR analysis of expression of CaSR in human spinal tissue. $n = 3$, Student's $t$-test.
E   Western blot analysis of expression of CaSR in human spinal tissue.
F   Immunofluorescence analysis of CaSR$^+$ (Green) and Runx2$^+$ (Red) cells in human spinal tissue.
G   Quantitative analysis of CaSR$^+$, Runx2$^+$ and CaSR$^+$ Runx2$^+$ cells number per area (mm$^2$) and CaSR$^+$ cell percentage. $n = 5$, Student's $t$-test.
H   Immunofluorescence analysis of CaSR$^+$ (Green) and OCN$^+$ (Red) cells in human spinal tissue.
I   Quantitative analysis of CaSR$^+$, OCN$^+$ and CaSR$^+$ OCN$^+$ cells number per area (mm$^2$). $n = 5$, Student's $t$-test.

Data information: SP: spinous process; IL: interspinous ligament; SL: supraspinous ligament; UIL: uncalcified interspinous ligament; TZ: transitional zone. Data shown as mean ± SD. **$P < 0.01$ compared between groups. Scale bar: 100 μm.

At 24 weeks of age, most CaSR$^+$ cells were accumulated at entheseal sites on dorsal surfaces and expressed OCN compared to baseline (Fig 3G and H).

A semi-Achilles tendon transection (SMTS) model was established to study pathological new bone formation due to the disruption of stress transmission at the posterior calcaneal tuberosity (PCT), as mechanical stress plays a prominent role in experimental AS (McClure, 1983; Jacques & McGonagle, 2014; Wang *et al*, 2018b). The results of μCT analysis showed that the entheseal bony projection gradually enlarged at 4 and 8 weeks compared to the sham-operated group (Fig EV1A). H&E and SOFG staining revealed that areas of both uncalcified fibrocartilage (UF) (with rounded chondrocytes morphology) and calcified fibrocartilage (CF) (with calcified extracellular matrix and separated from UF by a tidemark) of the Achilles tendon were increased 4 and 8 weeks after surgery (Fig EV1B–D) (Raspanti *et al*, 1996; Hibino *et al*, 2007). RT–qPCR analysis showed that expression of CaSR was increased in entheseal tissues in SMTS model (Fig EV1E). Immunofluorescence staining demonstrated that cells highly expressing CaSR were mostly Runx2$^+$ osteoblasts, and the number of CaSR$^+$ Runx2$^+$ cells increased in both UF and CF areas 4 and 8 weeks after surgery (Fig EV1F and G). At 4 and 8 weeks, a large amount CaSR$^+$ cells had accumulated at the UF and CF of the Achilles tendon and expressed OCN, indicating CaSR$^+$ mature osteoblasts were involved in the development of bony projections (Fig EV1H and I).

These results suggested that CaSR$^+$ osteoblasts might be involved in pathological new bone formation in AS animal models with different types of hypothetical pathogenesis (Vieira-Sousa *et al*, 2015).

**Systemic inhibition of CaSR suppresses the ankylosing phenotype in animal models of AS**

To validate the critical role of CaSR in pathological new bone formation, a selective CaSR antagonist, NPS-2143, was administered systemically during the process of pathological new bone formation in these animal models. Following treatment with NPS-2143 in PGIS, μCT analysis revealed that spinal ankylosis was ameliorated compared to the control group (DMSO administration) at 24 weeks (Fig 4A). H&E and SOFG staining demonstrated that spinal pathological new bone formation was attenuated after NPS-2143 treatment (Fig 4B). Meanwhile, histological scores in the NPS-2143 group were decreased at 16 and 24 weeks (Fig EV2A). The incidence of spinal ankylosis and pathological new bone formation was decreased compared to the control group (DMSO administration) at 24 weeks (Fig 4C and D). Similarly, following treatment with NPS-2143 in the DBA/1 model, μCT analysis showed that pathological new bone formation (red area) in the hind paws was suppressed at 20 weeks of age (Fig 4E). Furthermore, the incidence of ankle pathological new bone formation was decreased compared to the control group (DMSO administration) at 20 weeks (Fig 4F and G). H&E and SOFG staining revealed that pathological new bone formation on both plantar and dorsal surfaces was decreased at 20 weeks of age in response to NPS-2143 treatment (Fig 4H). However, the clinical severity score of the NPS-2143 group did not decrease compared to the DMSO group (Fig EV2B). Similarly, in the SMTS model treated with NPS-2143, μCT analysis demonstrated reduction of bony projections 8 weeks after surgery (Fig 4I). H&E and SOFG staining showed both UF and CF areas were decreased in SMTS mice in response to NPS-2143 treatment (Fig 4J–L).

It has been proposed that inhibition of CaSR may stimulate parathyroid hormone (PTH) secretion, which might have an anabolic effect on bone formation (Gowen *et al*, 2000). To verify the effects of PTH, plasma PTH levels were measured, and results showed that plasma PTH levels peaked 4 h after dosing with NPS-2143, gradually returning to baseline at 20 h in DBA/1, C57BL/6 and BALB/c mice (Fig EV2C).

These results strongly suggest that inhibition of CaSR suppresses pathological new bone formation in AS animal models due to different acknowledged types of hypothetical pathogenesis (Vieira-Sousa *et al*, 2015).

**Suppression of the ankylosing phenotype by CaSR antagonists might involve the process of endochondral ossification**

Since CaSR is also expressed in chondrocytes and plays a role in modulating chondrogenic differentiation and endochondral ossification, whether it affects the process of pathological new bone formation needed to be clarified. In our study, administration of a CaSR agonist (SR) failed to promote chondrogenic differentiation of murine chondrogenic ATDC5 cells. However, NPS-2143 suppressed chondrogenic differentiation, as well as the transcription of chondrogenic marker genes, including Sox9 and Col2a1 (Fig EV3A and B). These results suggest that the inhibitory effect of NPS-2143 in the ankylosing phenotype might involve inhibition of endochondral ossification.

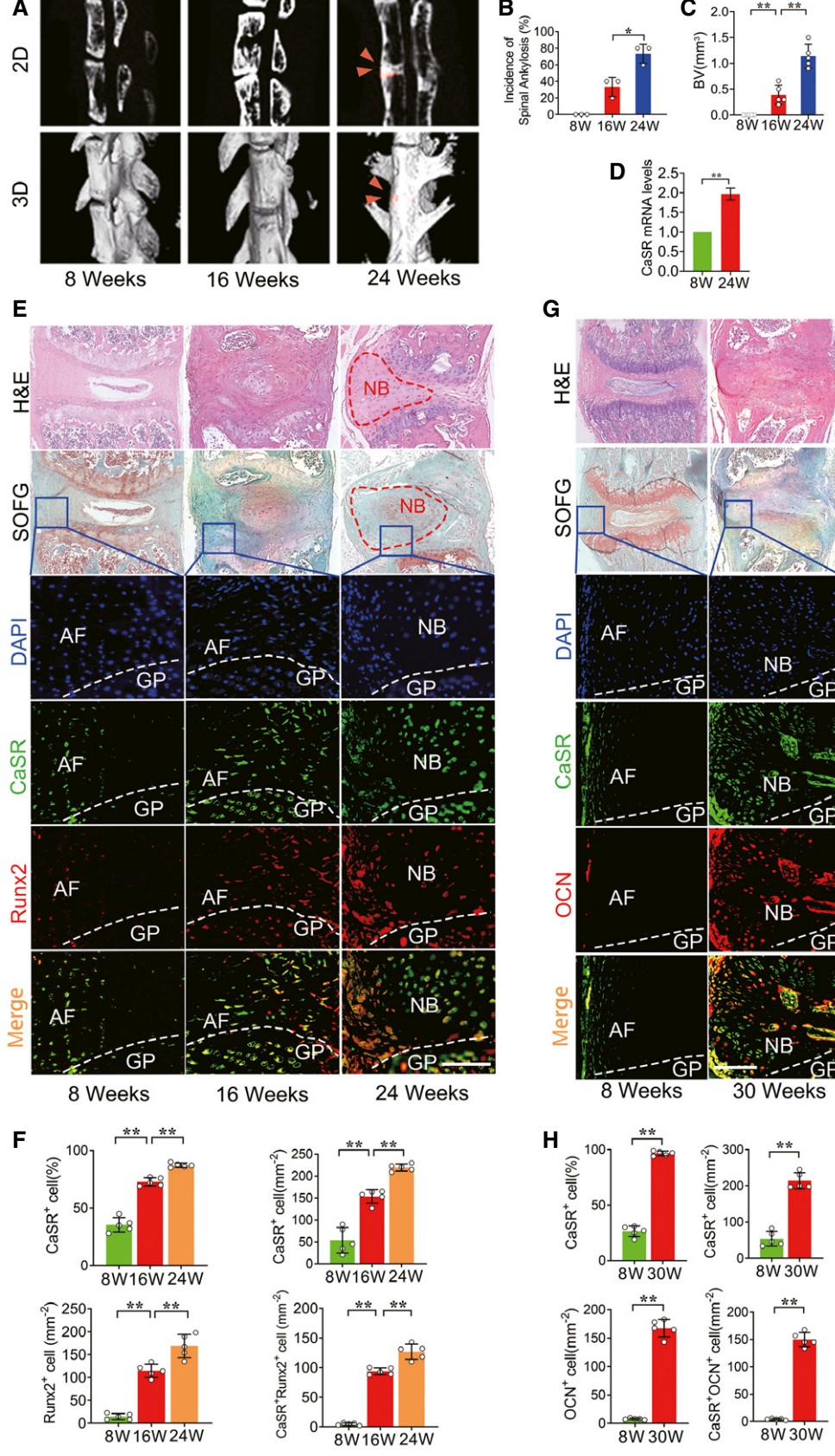

**Figure 2.**

**Figure 2. CaSR⁺ osteoblasts mediate new bone formation in PGIS model.**

A  μCT images of the spine in PGIS model (2D and 3D reconstruction). Arrow head shows spinal ankylosis. *n* = 5 per group. μCT images of new bone (red area) in hind paws (2D and 3D reconstruction) in DBA/1 model compared to baseline.
B  Incidence of spinal ankylosis in PGIS model. *n* = 5 per cage of total 3 cages per group, Fisher's exact test.
C  Quantitative analysis of structural parameters of spinal ankylosis by μCT analysis. *n* = 5 per group, one-way ANOVA, Bonferroni *post hoc*.
D  RT–qPCR analysis of expression of CaSR in spinal ligament tissue. *n* = 3, Student's *t*-test.
E  H&E, SOFG and Immunofluorescence analyses of the spine specimen of PGIS model compare to baseline. *n* = 5 per group.
F  Quantitative analysis of CaSR⁺, Runx2⁺ and CaSR⁺ Runx2⁺ cells number and CaSR⁺ cell percentage in the spine specimen of PGIS model. *n* = 5 per group, one-way ANOVA, Bonferroni *post hoc*.
G  H&E, SOFG and immunofluorescence analyses of the spine specimen of PGIS model at 30 weeks compared to baseline. GP: Growth plate. *n* = 5 per group.
H  Quantitative analysis of CaSR⁺, OCN⁺ and CaSR⁺ OCN⁺ cells number in the spine specimen of PGIS model. *n* = 5 per group, one-way ANOVA, Bonferroni *post hoc*.

Data information: AF: Annulus fibrosus; GP: Growth plate; NB: New bone. Data shown as mean ± SD. **$P < 0.01$ compared between groups. Scale bar: 100 μm.

## Activation of CaSR promotes osteogenic differentiation and pathological new bone formation through the PLCγ signalling pathway both *in vitro* and *in vivo*

To determine the role of CaSR in osteogenic differentiation of hBMSCs, cells were pre-treated with the CaSR-specific agonist, strontium ranelate (SR), alone or in combination with the CaSR antagonist NPS-2143 before osteogenic induction. Calcium deposition detected by Alizarin Red staining was increased in cells pre-treated with SR and decreased in those pre-treated with both SR and NPS-2143 under osteogenic induction (Fig 5A and B). Similar effects on the expression of osteogenic marker genes, including Runx2, Osx, ALP and OCN, were observed 24 h after osteogenic induction (Fig 5C). These results suggest that CaSR might be critical for osteogenic differentiation in hBMSCs.

To further determine whether the PLCγ signalling pathway was involved in CaSR-regulating osteogenic differentiation, hBMSCs were pre-treated with a PLCγ specific inhibitor (U73122) before SR stimulation under osteogenic induction. Calcium deposition was partially but suppressed in hBMSCs (Fig 5D and E). Similar effects on expression of the osteogenic markers Runx2, Osx, ALP and OCN were observed at mRNA levels after osteogenic induction (Fig 5F). In addition, Western blot analysis showed that SR increased the phosphorylation of PLCγ (activated), and NPS-2143 suppressed activation of PLCγ signalling by inhibiting CaSR (Fig 5G and H). These results suggest that CaSR promotes osteogenic differentiation through the PLCγ signalling pathway.

To confirm activation of the CaSR-PLCγ signalling pathway *in vivo*, we observed phosphorylation of PLCγ in the established animal models. In the PGIS model, immunofluorescence staining demonstrated that positive staining of p-PLCγ was notably increased in CaSR⁺ cells at the site of pathological new bone formation 24 weeks after induction, whereas NPS-2143 treatment leads to reduction of positive of p-PLCγ staining in CaSR⁺ cells (Fig 5I). In the SMTS model, immunofluorescence staining demonstrated that positive staining of p-PLCγ was increased in CaSR⁺ cells at the Achilles tendon (AT), UF and CF 8 weeks after surgery, and NPS-2143 treatment leads to reduced positive staining of p-PLCγ in CaSR⁺ cells (Fig 5J). A similar effect on PLCγ signalling inhibition was observed in DBA/1 models on dorsal surfaces following NPS-2143 treatment compared to activation of PLCγ signalling in the control group (Fig 5K). In the SMTS model administered U73122, μCT analysis revealed a reduction of bony projections 8 weeks after surgery (Fig 5 L). SOFG staining showed both UF and CF areas were

decreased in SMTS mice in response to U73122 treatment (Fig 5M and N). These results suggest that CaSR promotes osteogenic differentiation and pathological new bone formation through PLCγ signalling both *in vitro* and *in vivo*.

## Multiple inflammatory cytokines induce CaSR upregulation in osteoblasts rather than chondrocytes

Previous studies proved that multiple inflammatory cytokines, including IL-1β, TNFα, IL-17A, IL-22 and IL-23, are involved in pathological new bone formation in AS (van der Paardt *et al*, 2002; Sims *et al*, 2008; Sherlock *et al*, 2012; Tseng *et al*, 2016; Sieper & Poddubnyy, 2017; McGonagle *et al*, 2019; van Tok *et al*, 2019). Expression of IL-1β, TNFα, IL-17A, IL-22 and IL-23 was found increased in entheseal tissues from PGIS, DBA/1 and SMTS models (Fig EV4A–C). To determine whether upregulation of CaSR was induced by inflammatory cytokines, the MC3T3-E1 pre-osteoblast cell line was treated with inflammatory cytokines, including IL-1β, TNFα, IL-17A, IL-22 and IL-23. Results revealed that all theses inflammatory cytokines efficiently induced CaSR expression, confirmed by RT–qPCR and Western blot (Fig 6A–E). However, stimulation with these cytokines failed to induce upregulation of CaSR at the mRNA level in ATDC5 cells, indicating these inflammatory cytokines might not be able to induce upregulation of CaSR in chondrocytes (Fig EV2C).

To clarify the effect of CaSR on osteogenic differentiation in the inflammatory microenvironment, MC3T3-E1 cells transfected with CaSR siRNA or control siRNA were cultured in osteogenic differentiation medium with or without low-dose TNFα stimulation. The results showed that the osteogenic differentiation of the cells was suppressed by siCaSR treatment (Fig EV4D and E).

To further determine whether classic downstream signalling pathways of inflammatory cytokines regulate CaSR expression, the NF-κB/p65 or JAK/Stat3 pathway was selectively inhibited by p65 or Stat3 target-specific siRNA before inflammatory cytokine treatment. The results showed that by inhibiting the NF-κB/p65 pathway, expression of CaSR in MC3T3-E1 cells with IL-1β, TNFα and IL-17A stimulation was partially but significantly decreased (Fig 6A–C). Inhibition of the JAK/Stat3 pathway caused expression of CaSR to decrease at mRNA and protein levels partially but significantly in MC3T3-E1 cells in response to TNFα, IL-17A, IL-22 and IL-23 stimulation (Fig 6B, C and E). Overexpression of CaSR in MC3T3-E1 cells promoted its osteogenic potential compared to negative control. Transfection with p65 or Stat3 siRNA did not affect the

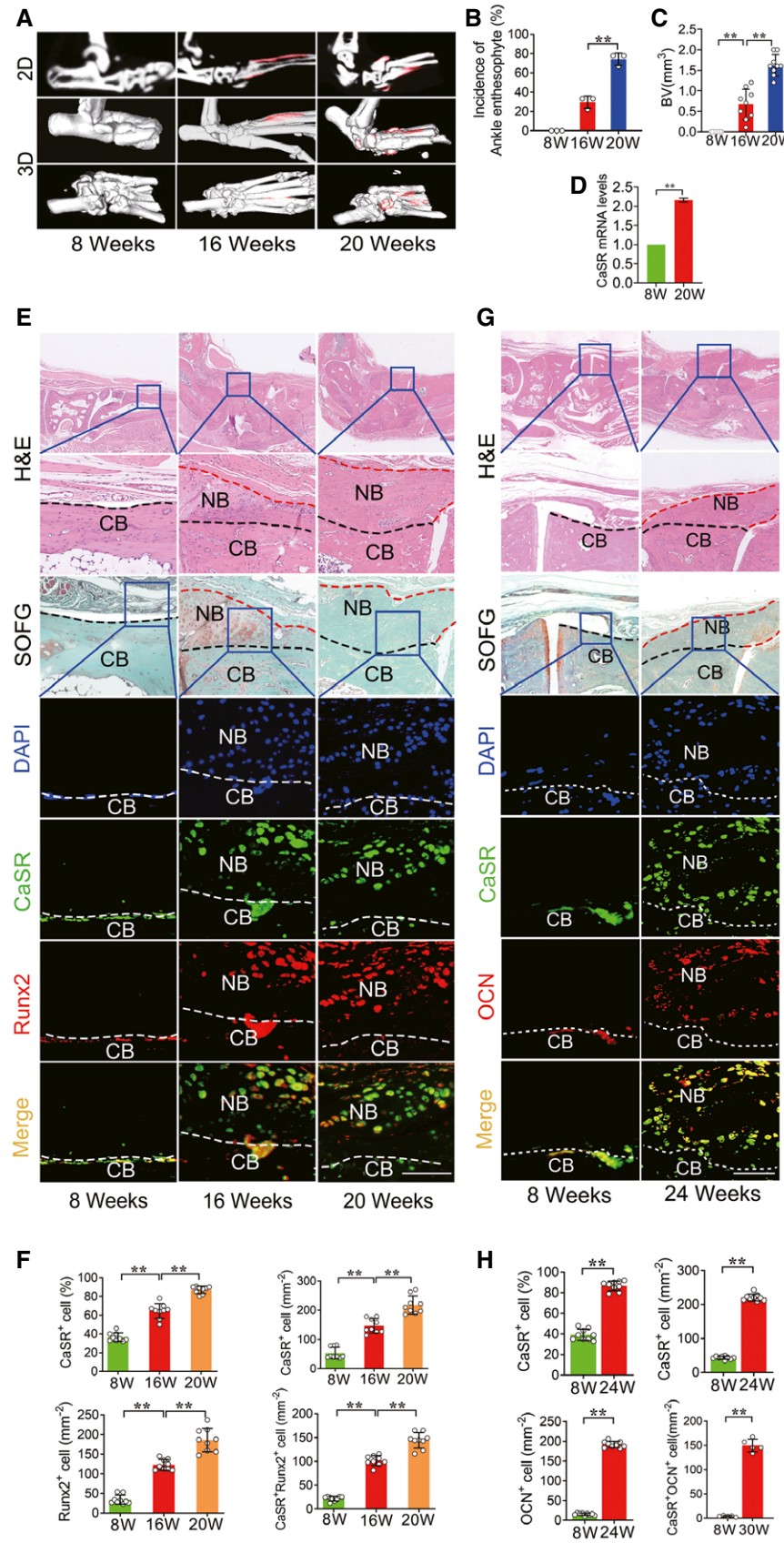

Figure 3.

◀

**Figure 3. CaSR+ osteoblasts mediate new bone formation in DBA/1 model.**

A  μCT images of pathological new bone formation in DBA/1 model. n = 9 per group.
B  Incidence of ankle pathological new bone formation in DBA/1 model. n = 9 per cage of total 3 cages per group, Fisher's exact test.
C  Quantitative analysis of structural parameters of pathological new bone by μCT analysis. n = 9 per group, one-way ANOVA, Bonferroni post hoc.
D  RT–qPCR analysis of expression of CaSR in hind paw tissue. n = 3, Student's t-test.
E  H&E, SOFG and immunofluorescence analyses of pathological new bone formation in hind paws. n = 9 per group.
F  Quantitative analysis of CaSR+, Runx2+ and CaSR+ Runx2+ cells number and CaSR+ cell percentage in pathological new bone formation sites. n = 9 per group, one-way ANOVA, Bonferroni post hoc.
G  H&E, SOFG and immunofluorescence analyses of pathological new bone in hind paws. n = 9 per group.
H  Quantitative analysis of CaSR+, OCN+ and CaSR+ OCN+ cells number in pathological new bone formation sites. n = 9 per group, one-way ANOVA, Bonferroni post hoc.

Data information: CB: Cortical bone; NB: New bone. Data shown as mean ± SD. **$P < 0.01$ compared between groups. Scale bar: 100 μm.

osteogenic effect of CaSR overexpression. These results indicate that the osteogenic effect of CaSR overexpression is independent of p65 or Stat3 pathways (Fig EV4F).

To validate whether TNFα and IL-17A have a synergistic effect in upregulating CaSR expression during the process of pathological new bone formation, MC3T3-E1 cells were treated with TNFα and IL-17A alone or combination. RT–qPCR analysis of CaSR expression showed that upregulation of combined TNFα and IL-17A was not significant compared to each treatment alone (Fig EV4G). These results suggest that multiple inflammatory cytokines induce CaSR upregulation in osteogenic precursor cells through NF-κB/p65 and JAK/Stat3 signalling.

To confirm the activation of these inflammatory signalling pathways in vivo, we further examined the phosphorylation of p65 and Stat3 in the established animal models. In the PGIS model, immunofluorescence staining demonstrated positive staining of p-p65 and p-Stat3 was increased in CaSR+ cells in spinal ligament tissues 16 and 24 weeks after PG induction (Fig 7A). Similarly, in the DBA/1 model, immunofluorescence staining demonstrated positive staining of p-p65 and p-Stat3 was increased in CaSR+ cells at pathological new bone formation sites on dorsal surfaces at 16 and 20 weeks of age (Fig 7B). In the SMTS model, immunofluorescence staining demonstrated positive staining of p-p65 and p-Stat3 were increased in CaSR+ cells located at the AT, UF and CF 8 weeks after surgery (Fig 7C). In addition, CD45+ leukocytes accumulated around Runx2+ cells in tissue samples collected frome AS patients and animal models during the process of pathological new bone formation (Fig EV4H–J). The proximity of CD45+ cells and Runx2+ cells provides the opportunity for communication between these two cell types.

These results suggest that multiple inflammatory cytokines induce CaSR upregulation through NF-κB/p65 and JAK/Stat3 signalling in osteogenic precursor cells to promote pathological new bone formation both in vitro and in vivo (Fig 7D).

# Discussion

Pathological new bone formation is a typical pathological feature in AS (Sieper & Poddubnyy, 2017). The correlation between inflammation and pathological new bone formation has so far not been clarified (Maksymowych et al, 2009; Pedersen et al, 2011; Lories & Dougados, 2012; Song et al, 2012; van der Heijde et al, 2012). Although it is acknowledged that inflammation induces secretion of osteogenic growth factors by immune cells and indirectly promotes

pathological new bone formation at the enthesis, it is still unknown whether it has any positive regulatory effect on osteogenic differentiation of precursor cells in the same pathological new bone-forming microenvironment (Ruiz-Heiland et al, 2012; Li et al, 2018).

CaSR plays an essential role in bone homeostasis in both neonatal and adult stages (Goltzman & Hendy, 2015). Previous studies have shown that CaSR is also highly involved in many inflammatory diseases (Chang et al, 2008; Dvorak-Ewell et al, 2011; Hendy & Canaff, 2016; Klein et al, 2016). However, whether it plays a role in pathological new bone formation in AS has not yet been reported. In the current study, we explored the role of CaSR in pathological new bone formation in AS and provided the first piece of evidence that CaSR expression was upregulated and CaSR + osteoblasts were accumulated in spinal ligament tissues collected from AS patients. These spinal ligament tissues were uncalcified tissues located right next to the calcified ligaments, which would potentially progress into calcified tissues and were considered bone-forming sites (Lories & Haroon, 2017; Bruijnen et al, 2018; Wang et al, 2018b). Our finding from human specimens from AS patients indicated that aberrant upregulation of CaSR might play a critical role in osteogenic differentiation of precursor cells and subsequent pathological new bone formation at these locations (Fig 1).

The diversity of AS phenotypes, sharing peripheral, axial and extra-articular manifestations (e.g. psoriasis, uveitis, and inflammatory bowel disease) with different degrees of severity, precludes the validity of using a single animal model for the study of human AS (Vieira-Sousa et al, 2015). To confirm that CaSR was involved in pathological new bone formation in vivo, three AS animal models with different types of hypothetical pathogenesis were established. PGIS model is a systemic autoimmune murine model in which the abundance of autoantigens in this microenvironment initiates self-sustaining immune reactions in susceptible strains that eventually lead to reactive new bone formation, ankylosis and spinal fusion (Bardos et al, 2005). So far, PGIS model is the unique reported model without genetically engineering that exhibits ankylosis phenotype in the spine to the best of our knowledge. Male DBA/1 mice spontaneously develop pathological new bone formation at entheses, which might be due to their susceptible genetic background, spontaneous inflammation, aggressive behaviour and sex-related hormones (Holmdahl et al, 1992; Matthys et al, 2003). This animal model is widely used for the investigation of entheseal new bone formation (Vieira-Sousa et al, 2015). In addition, mechanical loading has been recognized as an important factor that plays a critical role in enthesopathy and new bone formation, which is the typical pathological feather in AS (Jacques et al, 2014; Jacques &

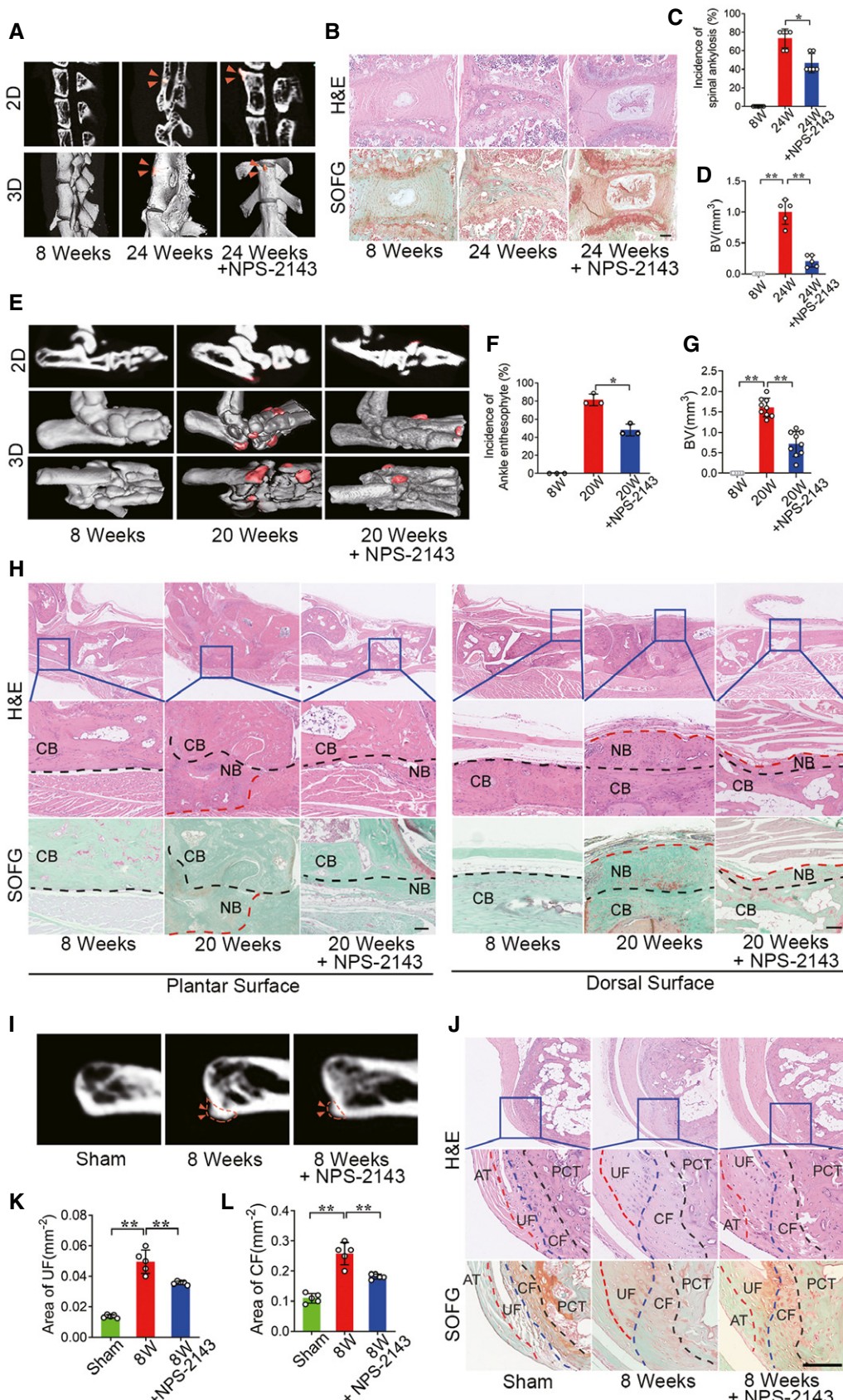

Figure 4.

**Figure 4. Systemic inhibition of CaSR suppresses the ankylosing phenotype of AS animal models.**

A  μCT images of the spine in PGIS model (2D and 3D reconstruction). Arrow head shows spinal ankylosis.
B  H&E and SOFG staining of spine in PGIS model.
C  Incidence of spinal ankylosis in PGIS model. $n = 5$ per cage of total 6 cages per group, Fisher's exact test.
D  Quantitative analysis of structural parameters of new bone by μCT analysis. $n = 5$, one-way ANOVA, Bonferroni *post hoc*.
E  μCT images of new bone formation (red area) in hind paws (2D and 3D reconstruction).
F  Incidence of new bone formation in DBA/1 model. $n = 9$ per cage of total 3 cages per group, Fisher's exact test.
G  Quantitative analysis of structural parameters of new bone by μCT analysis. $n = 9$, one-way ANOVA, Bonferroni *post hoc*.
H  H&E and SOFG staining of new bone in hind paws.
I  μCT images of the PCT in SMTS model (2D). Arrow heads shows bony projection.
J  H&E and SOFG staining of Achilles tendon enthesis compartment in SMTS model.
K  Quantitative analysis of area of UF. $n = 5$, one-way ANOVA, Bonferroni *post hoc*.
L  Quantitative analysis of area of CF. $n = 5$, one-way ANOVA, Bonferroni *post hoc*.

Data information: CB: Cortical bone; NB: New bone; UF: Uncalcified fibrocartilage; CF: Calcified fibrocartilage; PCT: posterior calcaneal tuberosity. Data shown as mean $\pm$ SD. **$P < 0.01$ compared between groups. Scale bar: 100 μm.

McGonagle, 2014; Ramiro *et al*, 2015; Schett *et al*, 2017; Wang *et al*, 2018b). It is hypothesized that mechanical stress and microdamage may be instrumental in inflammatory enthesitis as a primary driver (Jacques *et al*, 2014; Schett *et al*, 2017). To test whether aberrant upregulation of CaSR plays a critical role under this hypothesis, SMTS, a model with enthesopathy and pathological new bone formation due to unbalanced mechanical loading, was used (Wang *et al*, 2018b). Above all, our findings regarding CaSR from these animal models representing different hypotheses of pathogenesis of AS strongly suggested that aberrant upregulation of CaSR in osteoblasts is involved in pathological new bone formation in AS (Figs 1–3 and EV1).

To validate the critical role of CaSR in pathological new bone formation, a selective CaSR antagonist, NPS-2143, which binds to the seven transmembrane domains and disrupts G protein-mediated downstream signalling, was systemically administrated during the pathologic process of ankylosis in these animal models (Hannan *et al*, 2018b). Systemic administration of NPS-2143 reduced the incidence of pathological new bone formation and the severity of the ankylosing phenotype in animal models with different types of hypothetical pathogenesis (Fig 4). In addition, ablation of CaSR

suppressed osteogenic differentiation in the inflammatory environment (Fig EV4E). These findings indicate that CaSR plays a critical role in pathological new bone formation. Since it has been reported that NPS-2143 modulates the secretion of PTH via interaction with CaSR in the parathyroid gland, this might change skeletal homeostasis (Gowen *et al*, 2000). Therefore, plasma PTH levels of experimental animals were determined. Results showed that PTH levels were upregulated immediately after NPS-2134 application and sustained for approximately 6 h (twice the concentration compared to baseline), gradually reverting to baseline levels thereafter, in accordance with a previous report (Gowen *et al*, 2000). So far, no reported evidence has shown that systemic PTH levels are involved in pathological new bone formation in AS patients or animal models. Thus, reduced pathological new bone formation in response to NPS-2134 may not be related to its effect on PTH secretion or its systemic regulation of skeleton homeostasis (Fig EV2C).

Previous studies and the current study suggest that both endochondral ossification and intramembrane ossification are important in pathological new bone formation in AS. CaSR plays a critical role in modulating chondrogenic differentiation, chondrocyte maturation and subsequent endochondral ossification

**Figure 5. CaSR promotes osteogenic differentiation and new bone formation through PLCγ signalling pathway.**

A  Alizarin Red staining of hBMSC cells for 11 days.
B  Quantification of Alizarin Red staining at 570 nm (A570). $n = 3$, one-way ANOVA, Bonferroni *post hoc*.
C  RT–qPCR analysis of osteogenesis markers in hBMSC cells. $n = 3$, one-way ANOVA, Bonferroni *post hoc*.
D  Alizarin Red staining of hBMSC cells.
E  Quantification of Alizarin Red staining at 570 nm (A570). $n = 3$, one-way ANOVA, Bonferroni *post hoc*.
F  RT–qPCR analysis of osteogenesis markers in hBMSC cells. $n = 3$, one-way ANOVA, Bonferroni *post hoc*.
G  Western blot analysis of phosphorylation of PLCγ protein levels in hBMSC cells.
H  Western blot analysis of phosphorylation of PLCγ protein levels in hBMSC cells treated with SR and NPS-2143.
I  Immunofluorescence analysis of spine in PGIS model. Quantitative analysis of CaSR[+], p-PLCγ[+] and CaSR[+] p-PLCγ[+] cells number in spine of PGIS model. $n = 5$, one-way ANOVA, Bonferroni *post hoc*.
J  Immunofluorescence analysis of Achilles tendon enthesis compartment in SMTS model. Quantitative analysis of CaSR[+], p-PLCγ[+] and CaSR[+] p-PLCγ[+] cells number in Achilles tendon enthesis compartment. $n = 5$, one-way ANOVA, Bonferroni *post hoc*.
K  Immunofluorescence analysis of pathological new bone on dorsal surface of hind paws. Quantitative analysis of CaSR[+], p-PLCγ[+] and CaSR[+] p-PLCγ[+] cells number in pathological new bone formation sites. $n = 9$, one-way ANOVA, Bonferroni *post hoc*.
L  μCT images of the PCT in SMTS model (2D). Arrow heads shows bony projection.
M  SOFG staining of Achilles tendon enthesis compartment in SMTS model.
N  Quantitative analysis of area of UF and CF. $n = 5$, Student's *t*-test.

Data information: AF: Annulus fibrosus; CB: Cortical bone; NB: New bone; UF: Uncalcified fibrocartilage; CF: Calcified fibrocartilage; PCT: posterior calcaneal tuberosity. Data shown as mean $\pm$ SD. $n = 5$ for PGIS (I) and SMTS models (J); $n = 9$ for DBA/1 (K). *$P < 0.05$, **$P < 0.01$ compared between groups. Scale bar: 100 μm.

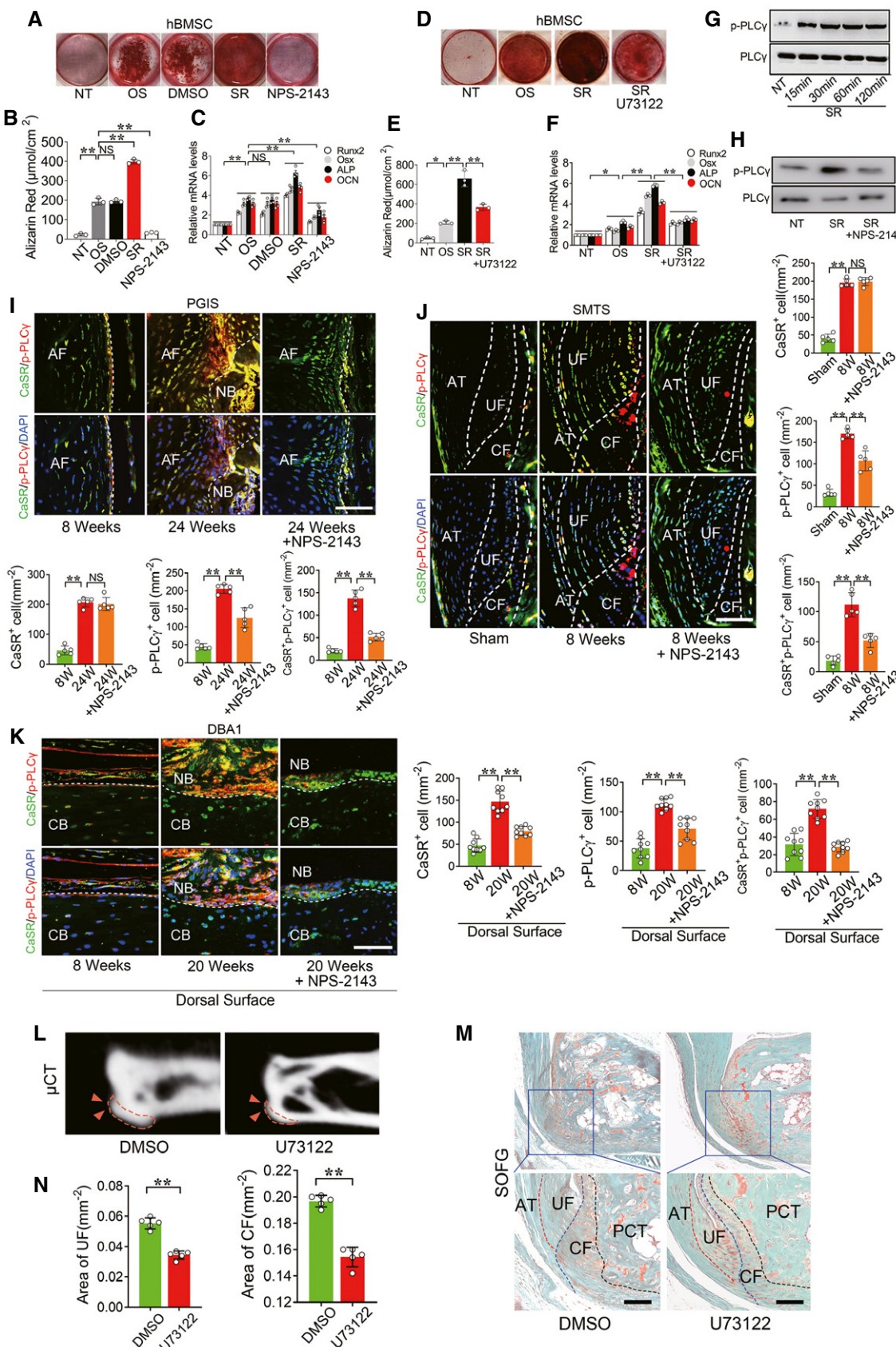

**Figure 5.**

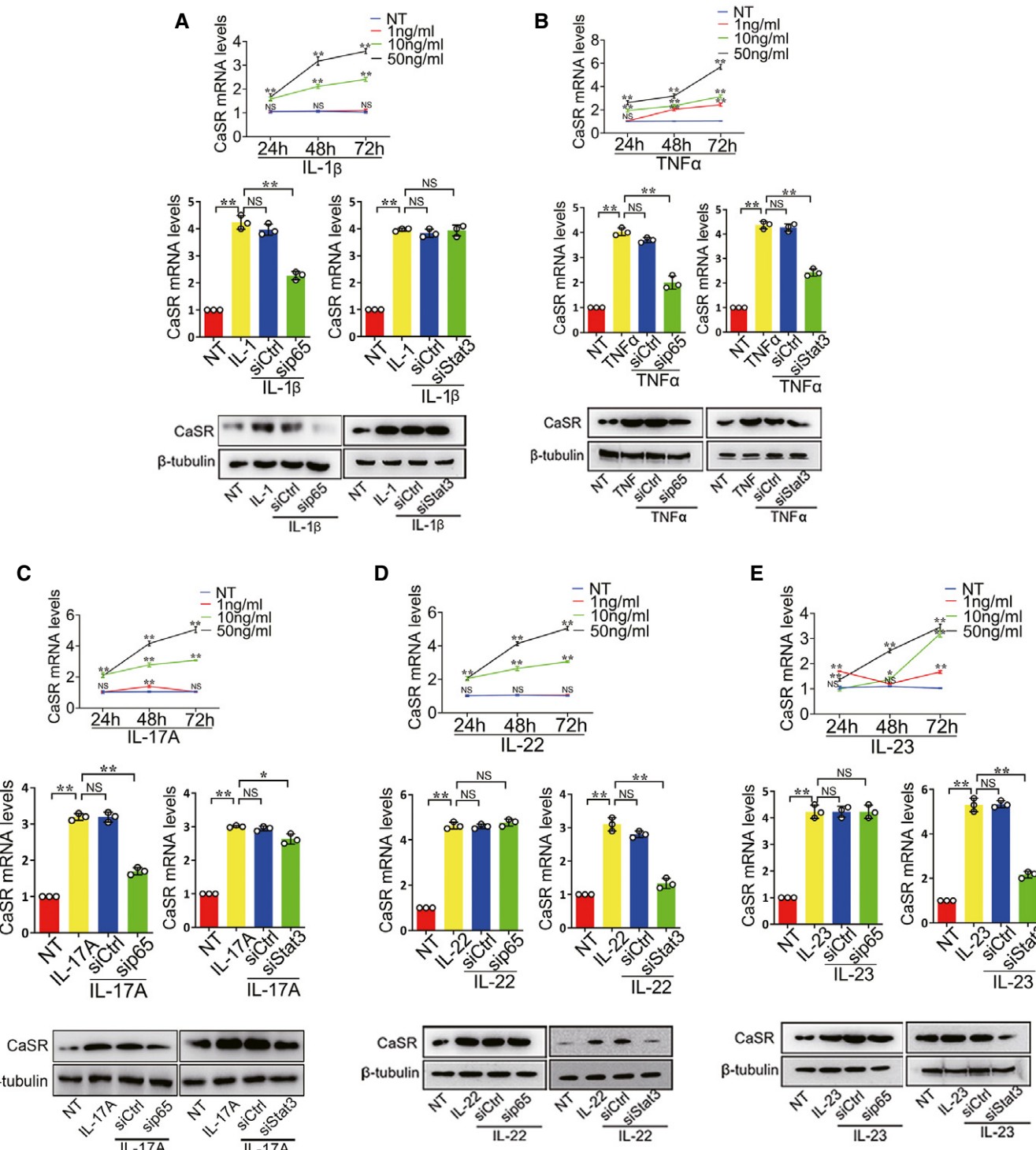

**Figure 6. Multiple inflammatory cytokines induce CaSR upregulation in osteogenic precursor cells.**

A  RT–qPCR and Western blot analyses of CaSR expressions in MC3T3-E1 cells treated with IL-1β. *n* = 3, one-way ANOVA, Bonferroni *post hoc*.

B  RT–qPCR and Western blot analyses of CaSR expressions in MC3T3-E1 cells treated with TNFα. *n* = 3, one-way ANOVA, Bonferroni *post hoc*.

C  RT–qPCR and Western blot analyses of CaSR expressions in MC3T3-E1 cells treated with IL-17A. *n* = 3, one-way ANOVA, Bonferroni *post hoc*.

D  RT–qPCR and Western blot analyses of CaSR expressions in MC3T3-E1 cells treated with IL-22. *n* = 3, one-way ANOVA, Bonferroni *post hoc*.

E  RT–qPCR and Western blot analyses of CaSR expressions in MC3T3-E1 cells treated with IL-23. *n* = 3, one-way ANOVA, Bonferroni *post hoc*.

Data information: The data are presented as the means ± SD from one representative experiment of three independent experiments performed in triplicate. #*P* < 0.05, **P* < 0.01 compared between groups.

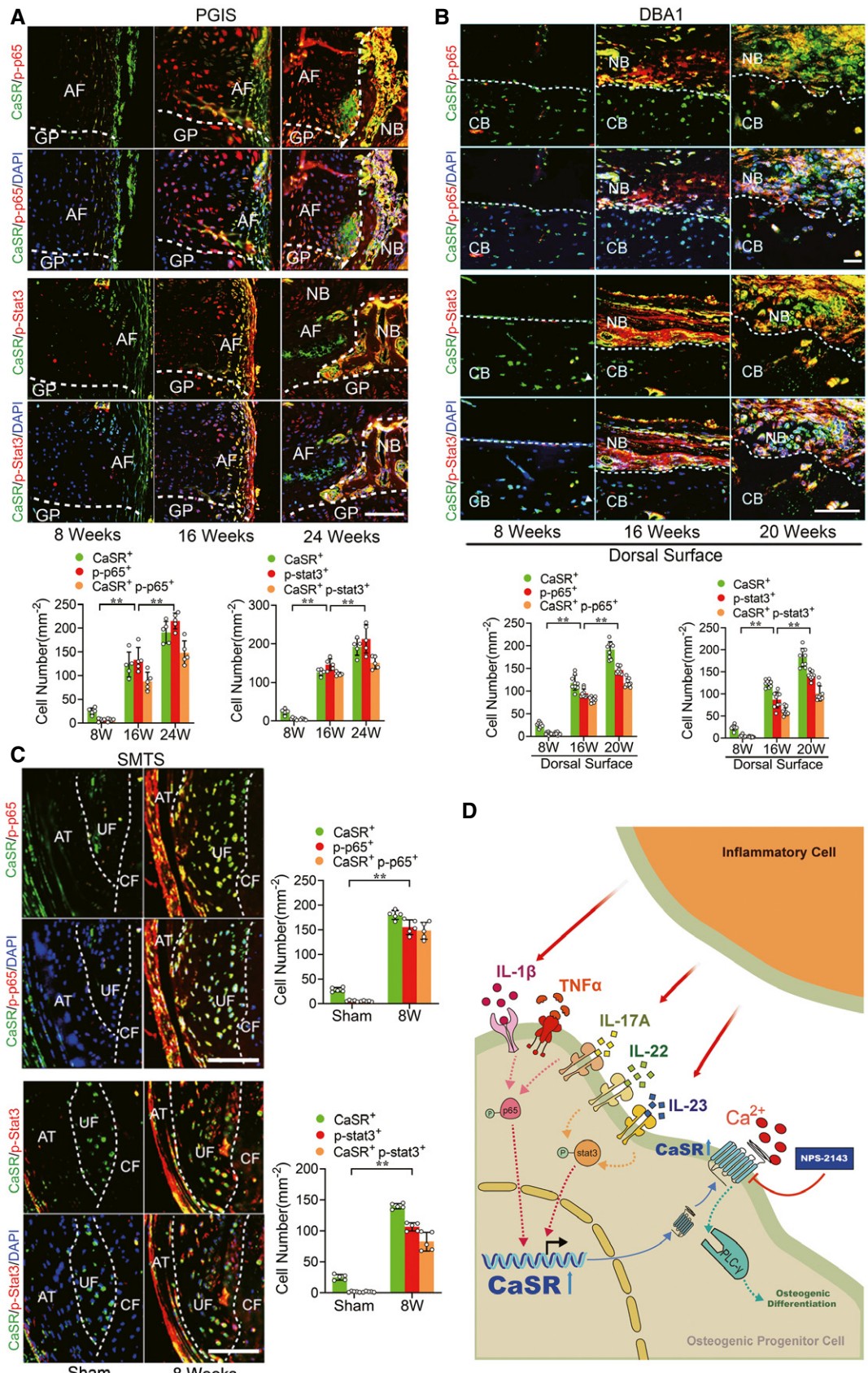

**Figure 7.**

◀

**Figure 7. Inflammatory signalling pathways are activated in osteogenic precursor cells in AS animal models.**

A  Immunofluorescence analysis of spine in PGIS mice. Quantitative analysis of CaSR$^+$ p-p65$^+$ cells and CaSR$^+$ p-Stat3$^+$ cells number in pathological new bone formation of PGIS model. $n = 5$, one-way ANOVA, Bonferroni *post hoc*.

B  Immunofluorescence analysis of pathological new bone of hind paws in DBA/1 mice. Quantitative analysis of CaSR$^+$ p-p65$^+$ cells and CaSR$^+$ p-Stat3$^+$ cells number in pathological new bone formation sites. $n = 9$, one-way ANOVA, Bonferroni *post hoc*.

C  Immunofluorescence analysis of Achilles tendon enthesis compartment in SMTS model. Quantitative analysis of CaSR$^+$ p-p65$^+$ cells and CaSR$^+$ p-stat3$^+$ cells number in PCT. $n = 5$, Student's *t*-test.

D  Schematic presentation of multiple inflammatory cytokines from inflammatory cell upregulate hyperstimulation of CaSR through NF-κB/p65 and JAK/Stat3 pathways. Hyperstimulation of CaSR induce osteogenic differentiation through PLCγ signalling pathway in osteogenic precursor cell to promote pathological new bone formation in AS.

Data information: AF: Annulus fibrosus; GP: Growth plate; CB: Cortical bone; NB: New bone; UF: Uncalcified fibrocartilage; CF: Calcified fibrocartilage; PCT: posterior calcaneal tuberosity. Data shown as mean $\pm$ SD. **$P < 0.01$ compared between groups. Scale bar: 100 μm.

(Goltzman & Hendy, 2015; Cheng *et al*, 2020). Knockdown of CaSR inhibited chondrogenesis of MSCs and hypertrophy and bone-forming matrix secretion of chondrocytes (Cheng *et al*, 2020). However, hyperstimulation of CaSR did not enhance chondrogenic differentiation of MSCs. These findings indicate that CaSR is essential but not sufficient for endochondral ossification (Sarem *et al*, 2018). Consistent with previous studies, our results showed that CaSR antagonist NPS-2143 suppressed chondrogenic differentiation of chondrogenic precursor cells. However, CaSR agonist SR failed to enhance chondrogenic differentiation in chondrogenic precursor cells (Fig EV3A and B). In addition, immune cytokines could upregulate expression of CaSR in osteoblasts but failed to increase the expression of CaSR in chondrocytes (Fig EV3C), indicating that inflammatory upregulation of CaSR mainly occurred in osteoblasts rather than chondrocytes during the process of pathological new bone formation. These results suggest that on one hand, inhibition of CaSR negatively impacted the process of endochondral ossification, meanwhile on the other hand directly suppressed intramembrane ossification.

Previous studies have noted that multiple downstream signalling pathways of CaSR participate in proliferation and osteogenic differentiation of osteoblasts, including PLCγ/PKC, MAPK (ERK1/2, JNK, p38) and cAMP/PKA (Yamaguchi *et al*, 2000; Godwin & Soltoff, 2002; Mizumachi *et al*, 2017). CaSR is reportedly linked to PLC via the heterotrimeric G protein isoforms Gq and mediates the downstream ERK1/2 pathway (Hannan *et al*, 2018a). PLC is a classic second massager of CaSR that is expressed in multiple organs, such as the parathyroid gland, vascular endothelium and vascular smooth muscle (Zhang *et al*, 2019; Ma *et al*, 2020). PLC protein mediates extracellular Ca$^{2+}$ signalling and downstream biological functions, including hormone secretion, apoptosis and wound repair (Milara *et al*, 2010; Hannan *et al*, 2018a; Ma *et al*, 2020). Moreover, the PLCγ/PKC signalling pathway mediates osteogenic differentiation in osteoblasts (Moenning *et al*, 2009). Our findings demonstrated that the PLCγ pathway is involved in pathological new bone formation. Activation of PLCγ and subsequent pathological new bone formation was suppressed by the CaSR antagonist, NPS-2143. These results suggest that the CaSR-PLCγ signalling pathway is critical for pathological new bone formation *in vivo* (Fig 5). The above findings indicate that osteoblasts at entheseal sites may possess higher differentiation potential in the inflammatory microenvironment. Consistent with our results, some literature has stated that hBMSCs collected from AS patients differentiate towards bone nodules more efficiently than their non-AS counterparts under osteogenic stimulation *in vitro* (Xie *et al*, 2016).

Another important finding in the current study is that multiple inflammatory cytokines, including TNFα, IL-1β, IL-17A, IL-22 and IL-23, promoted upregulation of CaSR in cultured osteoblasts in an independent manner. CD45$^+$ immune cells accumulated around Runx2$^+$ osteoblasts during the process of pathological new bone formation, providing an opportunity for communication between immune cells and osteoblasts (Fig EV4H–J). IL-1β and IL-17A induced CaSR primarily through the NF-κB pathway, while IL-22 and IL-23 induced CaSR primarily through the JAK/Stat3 pathway. In addition, TNFα induced CaSR through both pathways (Figs 6 and 7). Recently, it has been reported that anti-IL-17 single application insufficiently reduced spinal pathological new bone formation in AS, while TNFi application during radiographic intervals in AS exerts minimal effect on spinal radiographic progression despite its clinically relevant inhibitory effect on spinal radiographic changes if applied before radiographic intervals (Sieper & Poddubnyy, 2016; Deodhar, 2018; Molnar *et al*, 2018; van Tok *et al*, 2019). Therefore, aberrant expression of CaSR independently induced by different inflammatory cytokines might partially explain the unsatisfying outcomes of either TNFi or anti-IL-17 administration. In the current study, although the CaSR antagonist did not suppress inflammation *in vivo* (Fig EV3B), it directly suppressed pathological new bone formation in animal models (Fig 4). These results indicate that suppression of CaSR and the involved downstream pathways might be more useful to the sufficient therapy for pathological new bone formation in AS.

There are certain limitations in the current study. First, although the focus of the current study was the aberrant expression of CaSR in osteoblasts, CaSR is also expressed in Runx2$^-$ OCN$^-$ cells in the pathological new bone-forming area, and its role in other cell types in the pathological process remains unknown. An AS animal model established under the background of osteoblast-specific ablation of CaSR should be the best method to specifically verify the effects of CaSR on pathological new bone formation. However, since the genetically modified animal models currently used are mostly established under the C57BL/6 background, some strains, such as BALB/c and DBA/1 mice, require backcrossing with CaSR$^{flox/flox}$ mice to establish genetically modified AS models for their own specific genetic background, which is considered critical for disease onset and progression. The biological effect of CaSR specifically in osteoblasts will be confirmed in our future work. Second, as CaSR is activated by extracellular Ca$^{2+}$, the disadvantage of lacking a technique to detect dynamic changes in extracellular Ca$^{2+}$ near pathological new bone formation sites needs further investigation. Third, CaSR is also reported to be involved in many inflammatory diseases

(Hendy & Canaff, 2016; Klein *et al*, 2016). Whether augmentation or maintenance of inflammation by CaSR has a role in AS needs further investigations.

In summary, the novel findings of the current study suggest that inflammation-induced aberrant upregulation of CaSR and activation of CaSR-PLCγ-signalling in osteogenic precursor cells acts as a mediator of inflammation affecting pathological new bone formation in AS. Targeting CaSR may be a novel potential therapeutic strategy to slow the progression of axial structural ankylosis in AS.

# Materials and Methods

### Patients

Forty-two patients (18 with AS and 24 with non-AS) were consecutively enrolled at two hospitals between September 2013 and June 2019. AS patients who failed maintain a horizontal gaze in a natural, erect posture with the hips and knees comfortably extended fulfilled the criteria for deformity correction (Sciubba *et al*, 2008). AS patients with other diseases such as trauma, tumour, severe osteoarthritis in the lower limbs and a history of spine surgery were excluded from the study. Non-AS patients fulfilled the criteria for correction of scoliosis or spinal decompression of thoracic or lumbar vertebrae (Lenke *et al*, 2001; Kreiner *et al*, 2013). Non-AS patients were not known or suspected to have any systemic inflammatory condition including SpA (Appendix Table S1). In order to observe the molecular changes in the tissues that would potentially turn into calcified tissues, human spinal tissues obtained during corrective surgeries were the uncalcified tissues at the intermitted ossification stage (purple) (Fig EV5A). The Medical Ethics Committee of the First Affiliated Hospital of Sun Yat-sen University approved the procedures performed in this study. Informed consent was obtained from all human subjects, and the experiments conformed to the principles set out in the WMA Declaration of Helsinki and the Department of Health and Human Services Belmont Report.

### Mice

Inbred, female BALB/c, male DBA/1 and C57BL/6J mice (8/9 weeks old) weighing between 18–22 g were purchased from the Charles River Laboratories. BALB/c, DBA/1 and C57BL/6J mice always housed on a 12h light/dark cycle with unrestricted access to food and water. Mice were randomly separated into each experiment group. For the PGIS model, BALB/c mice were mixed and caged in groups of 5 mice at 24 weeks of age. Cartilage PG was prepared as previous described (Mikecz *et al*, 1987; Tseng *et al*, 2016; Tseng *et al*, 2017). As a standard method, the first antigen injection (100 mg PG protein) was given in complete Freund's adjuvant (Sigma, MO, USA), and the same doses of antigen were injected as second and third boosts in incomplete Freund's adjuvant on weeks 3 and 6. For CaSR inhibition, the mice received treatment orally three times per week with NPS-2134 (80 μmol/kg) (Sigma, MO, USA) or DMSO as negative control 2 weeks after third boost. Mice were euthanized at 8, 16 and 24 weeks after immunization. Spine specimens were dissected and fixed in 4% paraformaldehyde for histological analysis (*n* = 5 per group).

For the semi-Achilles tendon transection (SMTS) model, C57BL/6J mice were semi-transected the Achilles tendon to generate a destabilized enthesopathy animal model adjusted as in a previously described procedure (McClure, 1983; Wang *et al*, 2018b). Briefly, 3-month-old male C57BL/6J mice were anesthetized by ketamine and xylazine followed by semi-transection of Achilles tendon in the middle of the tendon body to induce abnormal mechanical loading-associated enthesopathy on the left calcaneus. Sham operations were done on independent mice. For CaSR inhibition, the mice received treatment orally three times per week with NPS-2134 (80 μmol/kg) (Sigma, MO, USA) or DMSO as negative control 3 days after surgery. For PLCγ inhibition, the mice were intraperitoneally injected three times per week with 1 mg/kg U73122 (Sigma, MO, USA) or DMSO as negative control 3 days after surgery. Mice were euthanized at 4 and 8 weeks after immunization. Hind paw specimens were dissected and fixed with 4% paraformaldehyde for histological analysis (*n* = 5 per group).

For spontaneous-arthritis DBA model, male DBA/1 mice were mixed and caged together in groups of 9 mice to induce arthritis at the age of 8 weeks (Lories *et al*, 2004; Lories *et al*, 2005; Li *et al*, 2018). Both hind paws were evaluated, resulting in a maximum score of 8. For CaSR inhibition, the mice received treatment orally three times per week with NPS-2134 (80 μmol/kg) (Sigma, MO, USA) or DMSO as a negative control 2 weeks after caging. Mice were euthanized at the age of 8, 16 and 20 weeks. Hind paw specimens were dissected and fixed with 4% paraformaldehyde for histological analysis (*n* = 9 per group). Mice that have severe disease and resulted in> 15% body weight loss will be killed and excluded. In the current study, no mice reached the humane end points.

The Medical Ethics Committee of the First Affiliated Hospital of Sun Yat-sen University approved the procedures performed in this study.

### Evaluation of severity of arthritis

DBA/1 mice were scored twice a week for clinical signs of arthritis as follows: Clinical score was graded per paw as follows: 0, normal; 1, swelling of one digit; 2, swelling of two or more digits; or 3, swelling of the entire paw. Scores were evaluated in a blinded manner as described previously (Sherlock *et al*, 2012).

Severity of vertebral joint disease was scored as described previously (Bardos *et al*, 2005); score 1, enthesitis, inflammatory cell accumulation around the IVD and/or infiltration of the annulus fibrosus; score 2, < 50% absorption/erosion of the IVD; score 3, essentially complete resorption (> 50%) of the IVD; score 4, cartilaginous/bony ankylosis.

### Biochemical assays

The concentrations of PTH in plasma were measured using Mouse Parathyroid Hormone EIA Kit (RayBio).

### Flow cytometric analysis

For identification of hBMSCs, cells were incubated with the monoclonal antibodies: FITC-conjugated CD90 (1:50), FITC-conjugated CD105 (1:50), FITC-conjugated CD14 (1:50), FITC-conjugated CD45 (1:50) and FITC-conjugated CD73 (1:50). The cells were analysed by

flow cytometry using CytoFLEX (Beckman Coulter). Data were acquired as the fraction of labelled cells within a single-cell gate set of 10,000 events (Fig EV5B).

## Cell treatments

To induce osteogenesis, different progenitor cells were plated in a 6-well plate at a density of $2 \times 10^5$ cells/well and cultured for 24 h. The cells were then switched to osteogenic medium consisting of α-minimum essential medium supplemented with 10% foetal bovine serum (FBS), 50 μg/ml L-ascorbic acid, 0.1 μM dexamethasone and 10 mM b-glycerophosphate to induce osteogenesis. The medium was changed every 3–4 days. To assess inflammatory stimulation, 10 ng/ml TNFα was added in osteogenic medium.

To assess chondrogenic differentiation, 100,000 ATDC5 cells (murine chondrogenic cell line) (Riken Cell Bank (Tsukuba, Japan)) were resuspended in 10 μl of control medium and seeded as micro-masses in the middle of a 24-well plate. Cells could attach for 1 h at 37°C, after which 0.5 ml of control, DMEM containing 10 ng/ml recombinant mice TGFβ1 (PeproTech), 50 μM L-ascorbic acid 2-sulphate (Sigma-Aldrich) were added to the wells. Medium was refreshed every other day, and after 9 days, micromasses were either stained with alcian blue or used for RNA isolation.

To evaluate the effect of NPS-2143 during osteogenesis or chondrogenesis, NPS-2143 (0.1 μM) was added 1 h before strontium ranelate (SR, 1 mM) stimulation. To evaluate the effect of PLCγ pathway, U73122 (2 μM) was added 1 h before strontium ranelate (SR, 1 mM) stimulation. To investigate the effect of IL-1β, TNFα, IL-6, IL-17A, IL-22 and IL-23 on CaSR expression, 1, 10 and 50 ng/ml of these inflammatory cytokines were added on MC3T3-E1 cells. To investigate the effect of these inflammatory cytokines on CaSR expression in chondrocytes, 10 ng/ml of these inflammatory cytokines was added on ATDC5 cells.

## RNA extraction and quantitative real-time PCR

Tissue samples were flash-frozen in liquid nitrogen and stored at −80°C. Samples were homogenized separately in TRIzol (Life Technologies, Mulgrave, Victoria, Australia). For gene expression analysis, Total RNA was extracted from cells according to the manufacturer's protocol, and 2 μg of total DNA-free RNA was used to synthesize cDNA using the ReverTra Ace qPCR RT Kit (Toyobo, Osaka, Japan). The reactions were set up in 96-well plates using 1 μl cDNA with Thunderbird SYBR qPCR Mix (Toyobo, Osaka, Japan), to which gene-specific forward and reverse PCR primers were added. qRT–PCR was performed under the following conditions: 95°C for 10 min, followed by 40 cycles of 95°C for 10 s and 55°C for 34 s. Analysis was performed to detect CaSR, p65, Stat3, Runx2, Osx, ALP, OCN, Sox9 and Col2a1 expression, and β-actin was used as an internal control. Primer sequences were as Appendix Table S2.

## Western blot

Spinal tissue protein was extracted with T-PER tissue protein extraction reagent (Thermo Scientific), following the instructions of the manufacture. After protein extraction, protein concentration was determined with a BCA assay. A 10% SDS–PAGE gel was loaded with 20 μg of total protein, and the separated proteins were transferred by electro blotting to PVDF membranes. The membranes were blocked with 5% non-fat dry milk in TBST (50 mM Tris, pH 7.6, 150 mM NaCl, 0.1% Tween 20) and incubated with the primary antibody overnight at 4°C in 5% non-fat dry milk in TBST. Immunolabelling was detected using ECL reagent (Invitrogen, CA). The antibodies used for Western blot were from following source: anti-CaSR antibody (Abcam, UK; 1:1,000), anti-PLCγ antibody (Cell Signaling Technology, MA; 1:1,000), anti-phospho-PLCγ antibody (Cell Signaling Technology, MA; 1:1,000), anti-β-actin antibody (Sigma-Aldrich, MO; 1:10,000) and anti-β-tubulin antibody (Sigma-Aldrich, MO; 1:10,000).

## RNA interference

The small interfering RNA(siRNA) duplexes were pre-designed with the online software (Stealth RNAi Pre-Designed siRNAs) provided by Ambion (Thermo Scientific™) (http://www.thermofisher.com/cn/zh/home/life-science/rnai/synthetic-rnai-analysis/stealth-select-rnai.html) and constructed by GenePharma (GenePharma Co., Suzhou, China). siRNA duplexes were presented in Appendix Table S3. The siRNAs would be verified to be efficient before all experiments (Fig EV5C–E). Cells were plated at a concentration of $1 \times 10^5$ cells/well in 6-well plates and transduced with the small interfering RNA (siRNA) using lipofectamine RNAiMAX transfection reagent (Invitrogen, USA) according to the manufacturer's instructions. Different amounts of 20 μM siRNA duplexes were mixed with 5 μl/well of transfection reagent, and Opti-MEM reduces serum medium (Invitrogen, USA) to total volume of 500 μl and incubated for 20 min. The mixture was applied to cell 16 h at 37°C in 5% $CO_2$.

## Establishment of a stable CaSR-overexpression cell line

MC3T3-E1 cells were cultured in αMEM, and the plasmid carrying the CaSR gene and three helper plasmids were transferred into MC3T3-E1 cells by lentivirus infection, and the CaSR gene was inserted into the genome of the cells. GFP-positive cells were selected by flow cytometry. Monoclonal clones were selected to inoculate a 96-well plate. Positive clones were transferred to a 12-well plate to expand culture and appropriate CaSR overexpressing stable cell lines were selected according to the expression level. At the same time, cell lines with empty plasmids were constructed.

## Mineralization analysis

For detecting the mineralization, we used OS to induced MC3T3-E1 or hBMSCs for 11 days. Cells were washed three times with PBS and fixed with 70% ethanol for 10 min. After three washes with distilled water, the cells were stained with a 40 mM alizarin red S (Sigma-Aldrich) solution (pH 4.1) for 10 min to visualize matrix calcium deposition. The remaining dye was washed three times with distilled water, and the stained cells were photographed. For quantification, the calcium deposits were distained with 10% cetylpyridinium chloride in 10 mM sodium phosphate (pH 7.0), then the extracted stain was transferred to a 96-well plate, and the absorbance of the samples was measured at 570 nm using a microplate reader (Tecan, Salzburg, Austria) (Li et al, 2018; Li et al, 2019).

## Immunofluorescence

Tissue sections were fixed in 4% PFA for 30 min and permeabilized with 0.3% Triton X-100 for 30 min. Blocking was performed with 5% normal goat serum for 1 h. The tissue sections and the cells were incubated overnight at 4°C in the primary antibodies against following antigens: anti-CaSR antibody (Abcam, UK; 1:200), anti-Runx2 antibody (Abcam, UK; 1:200), anti-p-PLCγ antibody (Abcam, UK; 1:200), anti-p-p65 antibody (Abcam, UK; 1:200), anti-p-Stat3 antibody (Abcam, UK; 1:200), anti-CD45 antibody (Abcam, UK; 1:100) and anti-Col2a1 antibody (Abcam, UK; 1:100). After washing three times in PBS, the primary antibodies were probed with the secondary antibodies Alexa Fluor 594 goat anti-rabbit (1:500, Invitrogen, Camarillo, CA) and Alexa Fluor 488 goat anti-mouse (1:500, Invitrogen) for 1 h at room temperature. Finally, the coverslips were washed in PBS three times and mounted using Prolong Gold Antifade Reagent containing 4′-6-diamidino-2-phenylindole (DAPI) (Molecular Probes, Invitrogen). The targeted marker-positive cells in each visual field were counted under a fluorescence microscope (Carl Zeiss Axio Observer Z1, Zeiss, Oberkochen, Germany). Data were acquired and analysed in a double-blinded manner. The investigators who performed the analysis comprised a separate group from investigators who performed immunofluorescence staining, and the two separate groups worked without knowing the different treatments designed by the principle investigator.

## μCT and histologic analyses

All specimens were obtained from mice post-mortem and fixed with 4% paraformaldehyde. For μCT scanning, specimens were fitted in a cylindrical sample holder and scanned using a Scanco lCT40 scanner set to 55 kVp and 70 lA. For visualization, the segmented data were imported and reconstructed as three-dimensional images using MicroCT Ray V3.0 software (Scanco Medical).

For histologic analysis, specimens were decalcified in 0.5M EDTA (Sigma-Aldrich) at 4°C. Paraffin-embedded sections were stained with haematoxylin and eosin (H&E) and Safranin O-Fast Green (SOFG) to evaluate general structures and bone formation. Immunohistochemical analysis of the specimens was conducted using specific antibodies. Tissue sections were quantitated according to the number of positive cells in per area as previously described (Li *et al*, 2018; Wang *et al*, 2018a,b).

## Statistics

Data obtained from experiments in duplicate or triplicate and repeated at least three times were represented as mean ± SD. Data were acquired and analysed in a double-blinded manner. The investigators who performed the analysis comprised a separate group from investigators who performed experiments, and the two separate groups worked without knowing the different treatments designed by the principle investigator. The unpaired Student's *t*-test was used to compare two groups with Shapiro–Wilk test for normality test. One-way ANOVA was performed with Levene's test for homogeneity of variance, followed by the Bonferroni post hoc test based on the comparison to be made and the statistical indication of each test. Mauchly's sphericity test was used for sphericity test. For non-parametric data, differences between groups were

> ### The paper explained
>
> #### Problem
> Upregulation of CaSR was found in AS spinal samples and animal models. Whether CaSR plays a role in the process of pathological new bone formation in AS is unknown.
>
> #### Results
> The expression of CaSR was upregulated in Runx2[+] and OCN[+] osteoblasts accumulated in the potential bone-forming sites in the tissues from AS patients and animal models. Systemic administration of CaSR antagonist NPS-2143 attenuated pathological new bone formation in animal models. Activation of PLCγ signalling by CaSR promoted osteogenic differentiation and pathological new bone formation. In addition, various inflammatory cytokines including TNFα and interleukin family induced upregulation of CaSR through NF-κB/p65 and JAK/Stat3 signalling pathways in osteoblasts.
>
> #### Impact
> Inflammation-induced upregulation of CaSR acts as a mediator of inflammation affecting pathological new bone formation in AS. Targeting CaSR may be a novel potential therapeutic strategy to slow the progression of axial structural ankylosis.

evaluated with non-parametric Mann–Whitney U-test, and categorical and binary variables were tested by the Fisher exact test. The exact sample size and the number of independent experiments performed, description of the samples and statistical analyses done were also specified in the figure legends. Statistical significance was accepted at $P < 0.05$. All graphs were generated using Prism V.7 (GraphPad), and all statistical tests were performed using SPSS V.21 (IBM). For detailed exact *P*-value per figure panel, see Appendix Table S4.

## Data availability

This study includes no data deposited in external repositories.

## Ethics approval

Medical Ethics Committee of the First Affiliated Hospital of Sun Yat-sen University.

**Expanded View** for this article is available online.

### Acknowledgements
National Natural Science Foundation of China [Grant no. 81972039; 81772307], Department of Science and Technology of Guangdong Province [Grant no. 2017A050501016; 2016TQ03R667], and Guangzhou Science and Technology Innovation Commission [Grant no. 201610010103].

### Author contributions
XL, SC, and ZH contributed equally to this work. HL conceived the ideas for experimental designs. XL, SC, and ZH conducted the majority of the experiments, analysed data, and prepared the manuscript. ZH, HW, and DC conducted sample collection and performed statistical analysis. JW, DC, ZL, and GD provided critical suggestions and instructions for the project and

helped compose the manuscript. XL and SC provided μCT analysis. XL, SC, ZL, HC, and LL conducted the most animal experiments and performed analysis. KZ, ZZhe, and ZZha provided suggestions for the project. HL developed the concept, supervised the project, and conducted data analysis.

## Conflict of interest

The authors declare that they have no conflict of interest.

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
