## [Review Process File · EMBO Molecular Medicine]

Aberrant upregulation of CaSR promotes pathological new bone formation in ankylosing spondylitis

Xiang Li, Siwen Chen, Zaiying Hu, Dongying Chen, Jianru Wang, Zemin Li, Zihao Li, Haowen Cui, Guo Dai, Lei Liu, Haitao Wang, Kuibo Zhang, Zhaomin Zheng, Zhongping Zhan, and Hui Liu

DOI: [10.15252/emmm.202012109](https://doi.org/10.15252/emmm.202012109)

Corresponding authors: Hui Liu (liuhui58@mail.sysu.edu.cn)

Review Timeline:

Submission Date:	4th Feb 20
Editorial Decision:	10th Mar 20
Revision Received:	8th Sep 20
Editorial Decision:	29th Sep 20
Revision Received:	13th Oct 20
Accepted:	14th Oct 20

Editor: Zeljko Durdevic

Transaction Report:

10th Mar 2020

Dear Prof. Liu,

Thank you for the submission of your manuscript to EMBO Molecular Medicine. We have now heard back from the referees whom we asked to evaluate your manuscript. As you will see from the reports below, the referees acknowledge the interest of the study. However, they raise some concerns that should be addressed in a major revision of the present manuscript.

We would welcome the submission of a revised version within three to six months for further consideration. Addressing the reviewers' concerns in full will be necessary for further considering the manuscript in our journal. I would also like to suggest that you run your article by an English native speaker to improve the grammar and syntax of the manuscript in order that the important messages of your manuscript are adequately conveyed.

Acceptance of the manuscript will entail a second round of review. Please note that EMBO Molecular Medicine encourages a single round of revision only and therefore, acceptance or rejection of the manuscript will depend on the completeness of your responses included in the next, final version of the manuscript. For this reason, and to save you from any frustrations in the end, I would strongly advise against returning an incomplete revision.

***** Reviewer's comments *****

Referee #1 (Remarks for Author):

This manuscript addresses up-regulation of CaSR promotes ectopic bone formation in ankylosing spondylitis. The manuscript also suggests a mechanism that CaSR regulates osteoblast through PLC γ in AS. The study is novel but there are major concerns. The authors confusingly used ectopic bone, osteophyte and enthesophyte, which are fundamentally different. The authors did not provide solid evidence for ectopic bone formation in AS patients or mouse models. Usage of PGIS and DBA/1 mice as AS models also need justification. The effect of CaSR antagonist on "ectopic bone" is underexplored. The author did not mention any side effect of this antagonist. Finally, the English is largely inadequate. Thus, the manuscript needs significant editing.

1. The authors use ectopic bone in the title but osteophytes or enthesophytes in the manuscript. Ectopic bone and osteophytes are conceptually different. These need clarification.
2. In the spinal tissues from AS patients, the bony projection is enthesophyte rather than ectopic bone.

3. Enthesis between spinous process and IF is a structure with a transitional zone. It is important to know whether OCN+ cells were in calcified IF or uncalcified IF in Fig. 1. Specific staining, such as Von Kossa, Col II immunostaining, SOFG staining, etc. are needed to identify different tissues.
4. OCN+ cells were accumulated in enthesal tissues rather than ectopic bone in AS patients.
5. PGIS mouse model is an established model for enthesitis rather than AS. The pathomechanism of enthesitis in PGIS model may be significantly different from that in AS. Thus, using PGIS as a AS model needs justification.
6. The resolution of some μ CT images is very low.
7. No ectopic bone was noted in μ CT image (Fig.2A).
8. Please mark osteophyte in the Fig. 2D.
9. It is not convincing that the immunofluorescence staining of CaSR and OCN were 100% overlaid in Fig.1F, 2F, 3F. More representative pictures may be needed.
10. Osteophytes are exostoses that form along joint margins, typically intra-articular. How could ectopic cartilage develop to osteophyte (Fig.2D)?
11. In PGIS mouse model, 24W SOFG staining showed cartilage in the ligament in Fig. 2D and ossification of IVD in Fig. 4B. Confusingly, IVD was intact at 30W in Fig. 2F.
12. It is reasonable to use SMTS model to study unbalanced mechanical force induced enthesopathy. However, SMTS model is not a well-accepted AS model, as AS is not caused by the imperfection of mechanical loading. Moreover, the results of SMTS models reflect enthesopathy other than ectopic bone formation. The manuscript lacks the rationale why SMTS model were used.
13. The spontaneously occurring arthritis in DBA/1 mice will develop enthesophyte formation, which is the hallmark of enthesopathy. This model is not suitable for study ectopic bone. How DBA/1 mice are related to AS also needs justification (such as genome wide association studies)
14. The baseline described in Fig.4C is very confusing. How treatment group is better than the baseline when there was no ankylosis or ectopic bone.
15. CaSR antagonist increased the PTH level to decrease OCN+ cells. However, the effect of PTH on bone is anabolic. The authors should explain why the ectopic bone formation were decreased after CaSR antagonist treatment? Were there any osteoclasts?
16. As the CaSR antagonist could regulate PTH level. Are there any side effect of CaSR antagonist?
17. The English is largely inadequate, and thus the manuscript needs significant editing.

Referee #2 (Comments on Novelty/Model System for Author):

The manuscript from Li et al. describes a study in which CaSR is overly expressed in the pre-matured and matured osteoblasts of entheses tissues obtained from AS patients as well as distinct AS animal models. The authors also use the CaSR antagonist NPS-2143 in the animal models to show that blocking CaSR prevents aberrant bone formation. In functional assays, the authors quantified the bone matrix (calcium deposition) in OS-cultured hBMSC with or without NPS-2134 treatment. They also observed that CaSR/PLCy axis promotes the bone formation through enhanced osteogenic differentiation. Furthermore, the authors determined that major AS-related inflammatory cytokines induce NF- κ B and/or JAK/STAT3 signaling pathways that promote the expression of CaSR in pre-osteoblast cell lines. The expression of p-p65 and p-STAT3 were upregulated in the areas of bone formation in the AS animal models.

Strong points:

- The manuscript contains novel experimental data using both human and animal samples.
- The imaging studies and histology assessments were nicely performed at different time points up to 24 weeks with appropriate sample numbers.
- The authors assessed various enthesis tissues from spine and peripheral joints, which suggested that CaSR is widely involved in the process of bone formation regardless of anatomical differences.
- The manuscript is well written.

Major concerns:

Given that enthesal tissues give rise to bone formation through the process of endochondral ossification, osteo-chondro progenitor cells first differentiate into chondrocytes and subsequently differentiate into osteoblast-making bone matrix. The authors also described the importance of endochondral ossification as a pathological process of ectopic bone formation in the Discussion. However, the authors mainly focus on osteoblast differentiation. As chondrocytes also highly express CaSR (Fig 1-3), it would be interesting to see the effect of CaSR on chondrocytes differentiation from progenitor cells. If CaSR induces the differentiation into chondrocytes, it would suggest that CaSR is involved in the whole process of endochondral ossification. Otherwise bone formation may occur through intramembrane ossification rather than endochondral ossification.

Figure 1-3: Although AS patients and AS animal models have a significant population of CaSR-positive cells, CaSR is physiologically wide-expressed in various tissues to regulate important cellular functions such as maintaining systemic calcium homeostasis. The difference in the positive ratio between AS and healthy controls may be attributed to the difference in the total cell number of the tissues, rather than high positive ratio. Instead of providing the number of positive cells, percentages of positive cells would be ideal to show the upregulated CaSR. Also, providing a negative control would be helpful. Furthermore, as enthesitis lesions have significant immune cell infiltration, it would be informative to characterize the immune cell population with representative markers (such as CD3, CD19, and CD68) to distinguish the osteo-progenitor cells from immune cells in the histology slides in IHC or IF.

Fig2B: There is a difference in the percentage of spinal ankylosis depending on cages. Does the degree of inflammation make the difference? Or could this reflect different microbiota? I would suggest providing the correlation between the severity of inflammation and percentage of bone formation in the spine.

Fig3 A, B: I would suggest using the term "enthesophytes" instead of "osteophytes". The term of "osteophytes" give readers the impression of osteoarthritis rather than spondyloarthritis. Based on Fig 3B and 3D, it seems that all bone formation is initiated from the areas of enthesal tissues.

Therefore, enthesophytes would be a precise term to describe the ectopic bone formation.

Fig3. Again, it would be ideal to provide the correlation between the severity of inflammation and bone formation.

FigEV2: Given the inflammation severity is decreased in the group treated with NPS-2143 (CaSR antagonist) compared to controls, NPS-2143 seems to have a potential to suppress inflammation in the tissue. This suggests that less percentages of bone formation may be due to less inflammation compared to control. Fig 6 also shows that various inflammatory cytokines increase the expression of CaSR. I would suggest performing an experiment to test the direct effect of CaSR on bone formation or osteophyte differentiation under non-inflammatory conditions. A potential experiment would be to compare bone formation makers (RUNX2/OCN) between CaSR-knockout and WT cells with or without inflammatory condition. Alternatively, the authors could check the bone formation markers in CaSR-overexpressed MC3T3-E1 cells treated with stat3 and pp65 siRNAs.

Fig 6: Based on the figure, it seems that general inflammation increases the expression of CaSR in osteoblasts through NF- κ B and JAK/STAT3 signaling pathways regardless of the type of cytokine. The authors described that these cytokines are "AS-related inflammatory cytokines", which could be challenged in terms of sensitivity and specificity. There was no screening test performed to detect the cytokines. In addition, those cytokines are also strongly related to other forms of inflammatory arthritis such as rheumatoid arthritis. Although NF- κ B and JAK/STAT3 signaling are indispensable pathways in RA, RA has bone erosive changes unlike AS and lacks osteoproliferation. It is possible that CaSR is upregulated not only osteoblasts but osteoclasts or any other cells. I would suggest testing the expression of CaSR in bone erosion areas to see the activation of osteoclasts or immune cells. It is very important point to see the specific role of CaSR in AS. As DBA/1 arthritis model is also used for the model of RA, the authors will be able to address this question.

Referee #3 (Remarks for Author):

This very interesting manuscript reports a potentially important mechanism of new bone formation in AS. The effects of different cytokines illustrates how this could be related to inflammatory immune mediated stimuli from several relevant cytokines. A very important synergy in this context is that between TNF- α and IL-17A. It would be very informative to test if this synergy is found in activation of the CaSR pathway.

Point-By-Point Responses to the Reviewers' comments**Reviewer 1**

1. The authors use ectopic bone in the title but osteophytes or enthesophytes in the manuscript. Ectopic bone and osteophytes are conceptually different. These need clarification.

Response:

Thank you for the careful review and kind suggestion. We used the term of “ectopic bone formation” is based on the understanding that “ectopic bone” is ossification of tissues outside their usual origins in soft tissues such as muscle, subcutaneous tissue, and fibrous tissue adjacent to joints [1]. However, it indeed caused confusion under the reminder from this respected reviewer. We again go through the existing literatures on this research field and found that previous studies preferred to use the term of “new bone formation” in PGIS model and “enthesophyte” in DBA/1 and SMTS models [2-8]. Since multiple animal models with different types of hypothetical pathogenesis of AS were used in the current study, we intend to replaced “ectopic bone” with “pathological new bone” to generally describe the new bone formation in these animal models.

New bone formation in AS patients is usually described as “syndesmophyte” in literatures, which means osseous excrescences or bony outgrowths from the spinal ligaments as they attach to adjacent vertebral bodies [2,3]. Thus, new bone formation in AS patients was also described as “syndesmophyte” in the current study.

References for this response

1. Scott MA, Levi B, Askarinam A, Nguyen A, Rackohn T, Ting K, et al. Brief review of models of ectopic bone formation. *Stem Cells Dev.* 2012 Mar 20; 21(5):655-667.
2. van der Heijde D, Machado P, Braun J, Hermann KGA, Baraliakos X, Hsu B, et al. MRI inflammation at the vertebral unit only marginally predicts new syndesmophyte formation: a multilevel analysis in patients with ankylosing spondylitis. *Annals of the Rheumatic Diseases.* 2012 Mar; 71(3):369-373.
3. Sieper J, Poddubnyy D. Axial spondyloarthritis. *Lancet.* 2017 Jul 1; 390(10089):73-84.
4. Benjamin M, McGonagle D. The anatomical basis for disease localisation in seronegative spondyloarthropathy at entheses and related sites. *J Anat.* 2001 Nov; 199(Pt 5):503-526.
5. Benjamin M, McGonagle D. Histopathologic changes at "synovio-enthesal complexes" suggesting a novel mechanism for synovitis in osteoarthritis and spondylarthritis. *Arthritis Rheum-U.S.* 2007 Nov; 56(11):3601-3609.
6. Glant TT, Radacs M, Nagyri G, Olasz K, Laszlo A, Boldizsar F, et al. Proteoglycan-induced arthritis and recombinant human proteoglycan aggrecan G1 domain-induced arthritis in BALB/c mice resembling two subtypes of rheumatoid arthritis. *Arthritis Rheum.* 2011 May; 63(5):1312-1321.
7. Bardos T, Szabo Z, Czipri M, Vermes C, Tunyogi-Csapo M, Urban RM, et al. A longitudinal study on an autoimmune murine model of ankylosing spondylitis. *Ann Rheum Dis.* 2005 Jul; 64(7):981-987.
8. Lories RJ. Animal models of spondyloarthritis. *Curr Opin Rheumatol.* 2006 Jul; 18(4):342-346.

2. In the spinal tissues from AS patients, the bony projection is enthesophyte rather than ectopic bone.

Response:

Thank you for the careful review and we are sorry for this confusion. We have replaced “ectopic bone” with “pathological new bone” and “syndesmophyte” to describe the new bone in the spinal tissues from AS patients.

3. Enthesis between spinous process and IF is a structure with a transitional zone. It is

important to know whether OCN+ cells were in calcified IF or uncalcified IF in Fig. 1. Specific staining, such as Von Kossa, Col II immunostaining, SOFG staining, etc. are needed to identify different tissues.

Response:

Thank you for the careful review and kind suggestion. In non-AS patient, the transitional zone is recognizable. However, it is replaced by infiltration of fibrous tissue or pathological new bone in AS patients and become distinguishable [1, 2]. Since AS is a chronic inflammatory disease, different stages of the pathological process of new bone formation can be observed at different locations of a single patients. The CT scanning showed that in position 1 (red) there was no ossification, with clearly recognizable margin of spinous process and interspinous ligament. Position 2 (blue) represented an intermitted ossification stage. Ossification of ligament was evident and the margin of spinous process and interspinous ligament was indistinct. Position 3 (green) represented a complete bony fusion stage, with spinous process and interspinous ligament completely fused (Figure EV5E). Thus, in the current study, in order to observe the molecular changes in the tissues that would be likely to turn into calcified tissues, human spinal tissues we obtained during corrective surgeries were the uncalcified tissues at the intermitted ossification stage (purple).

Figure EV5E

Figure EV5E. CT scanning of early (red), intermitted (blue), and late stage (green) of new bone formation in AS. SP: spinous process; IL: interspinous ligament; SL: supraspinous ligament; CIL: calcified interspinous ligament; UIL: uncalcified interspinous ligament.

In our results, SOFG staining showed that spinous process and calcified ligament is indistinguishable in AS. Meanwhile, increased cells accumulated at uncalcified zone in AS group (Figure 1H). Immunofluorescence staining of the same region showed that OCN⁺ cells were mainly accumulated at the uncalcified region, indicating that the uncalcified region might potentially develop into calcified region and lately to new bone.

The relative information has been added into revised Figure 1H and result section of revised manuscript (Page 7, Line 12-18; Page 17 Line 15-22).

Figure 1H

(**Figure 1H**) SOFG staining and Immunofluorescence analysis of CaSR⁺ (Green) and OCN⁺ (Red) cells in human spinal tissue. SP: spinous process; TZ: transitional zone; UIL: uncalcified interspinous ligament.

Reference for this response:

1. Benjamin M, McGonagle D. The anatomical basis for disease localisation in seronegative spondyloarthropathy at entheses and related sites. *J Anat.* 2001 Nov; 199(Pt 5):503-526.
2. Benjamin M, McGonagle D. Histopathologic changes at "synovio-enthesal complexes" suggesting a novel mechanism for synovitis in osteoarthritis and spondylarthritis. *Arthritis Rheum-U.S.* 2007 Nov; 56(11):3601-3609.

4. OCN+ cells were accumulated in enthesal tissues rather than ectopic bone in AS patients.

Response:

Thank you for the careful review and kind suggestion. We have changed the description as suggested. (Page 7, Line 14-18).

5. PGIS mouse model is an established model for enthesitis rather than AS. The pathomechanism of enthesitis in PGIS model may be significantly different from that in AS. Thus, using PGIS as a AS model needs justification.

Response:

Thank you for the careful review. Indeed, the model of immunization of susceptible mice (BALB/c strain) with human cartilage PG was firstly characterized as rapid development of a clinically and histologically demonstrable polyarthritis and ankylosing spondylitis [1]. This model has been used to study pathological process of rheumatoid arthritis and ankylosing spondylitis [2, 3]. Lately, it was found that some strains of BALB/c hybrids were susceptible to spondylitis without peripheral arthritis, while some strains were susceptible to peripheral arthritis without spondylitis. These results indicate that PG-induced arthritis and spondylitis might be independent diseases regarding to distinct genetic control of joint and spine involvement [3]. To our best knowledge, PGIS model was reported to be the only systemic autoimmune murine model which has typical spondylitis and axial new bone formation [3]. Glant T. et al proved that PG immunized BALB/c mice develop spondyloarthropathy (proteoglycan aggrecan-induced spondylitis (PGIS)), and the phenotype of the disease is very similar to human AS [3, 4]. Previous studies of PGIS model demonstrated

pathological phenotypes including intervertebral disc (IVD) destabilization, cartilage damage, chondrophyte/osteophyte formation, and their subsequent fusion [5]. Thus, PGIS model become an animal model to investigate the pathological process of enthesitis and new bone formation in the spine [3, 5-8].

This relative information has been added into discussion section of revised manuscript (Page 18, Line 4-14).

Reference for this response:

1. Glant TT, Mikecz K, Arzoumanian A, Poole AR. Proteoglycan-induced arthritis in BALB/c mice. Clinical features and histopathology. *Arthritis Rheum.* 1987 Feb; 30(2):201-212.
2. Glant TT, Radacs M, Nagyeri G, Olasz K, Laszlo A, Boldizsar F, et al. Proteoglycan-induced arthritis and recombinant human proteoglycan aggrecan G1 domain-induced arthritis in BALB/c mice resembling two subtypes of rheumatoid arthritis. *Arthritis Rheum.* 2011 May; 63(5):1312-1321.
3. Bardos T, Szabo Z, Czipri M, Vermes C, Tunyogi-Csapo M, Urban RM, et al. A longitudinal study on an autoimmune murine model of ankylosing spondylitis. *Ann Rheum Dis.* 2005 Jul; 64(7):981-987.
4. Lories RJ. Animal models of spondyloarthritis. *Curr Opin Rheumatol.* 2006 Jul; 18(4):342-346.
5. Tseng HW, Pitt ME, Glant TT, McRae AF, Kenna TJ, Brown MA, et al. Inflammation-driven bone formation in a mouse model of ankylosing spondylitis: sequential not parallel processes. *Arthritis Res Ther.* 2016 Jan 29; 18:35.
6. Haynes KR, Pettit AR, Duan R, Tseng HW, Glant TT, Brown MA, et al. Excessive bone formation in a mouse model of ankylosing spondylitis is associated with decreases in Wnt pathway inhibitors. *Arthritis Res Ther.* 2012; 14(6).
7. Tseng HW, Glant TT, Brown MA, Kenna TJ, Thomas GP, Pettit AR. Early anti-inflammatory intervention ameliorates axial disease in the proteoglycan-induced spondylitis mouse model of ankylosing spondylitis. *BMC Musculoskelet Disord.* 2017 May 30; 18(1):228.
8. Szabo Z, Szanto S, Vegvari A, Szekanecz Z, Mikecz K, Glant TT. Genetic control of experimental spondylarthropathy. *Arthritis Rheum.* 2005 Aug; 52(8):2452-2460.

6. The resolution of some μ CT images is very low.

Response:

Thank you for the careful review. All μ CT images with low resolution have been replaced by a version with high resolution in the revised Figure. We hope the revised version of the images will meet the publication standard.

7. No ectopic bone was noted in μ CT image (Fig.2A).

Response:

Thank you for the careful review and we are sorry for this confusion. Pathological new bone formation has been pointed out by arrowheads and high-lighted in Fig.2A. The result of μ CT revealed that the intervertebral space became narrow at 16 weeks and fused at 24 weeks due to the pathological new bone formation.

Figure 2A

(Figure 2A) μ CT images of the spine in PGIS model (2D and 3D reconstruction). Arrow head shows spinal ankylosis.

8. Please mark osteophyte in the Fig. 2D.

Response:

Thank you for the careful review. Pathological new bone has been circled with red dotted line in this figure, which has been relabeled as Figure 2E in the revised manuscript.

Figure 2E

(Figure 2E) H&E and SOFG staining analyses of the spine specimen of PGIS model compare to baseline. n=5 per group.

9. It is not convincing that the immunofluorescence staining of CaSR and OCN were 100% overlaid in Fig.1F, 2F, 3F. More representative pictures may be needed.

Response:

Thank you for the careful review and we are sorry for this confusion. In this study, we did not intend to show that CaSR and OCN were 100% overlaid. Previous studies reported that CaSR is physiologically wide-expressed in multiple lineages of cells to regulate important cellular functions [1, 2]. Our results also indicated that CaSR⁺ cells at enthesal sites are mainly osteo-lineage cells of different osteogenic stages. At early stage, increased CaSR⁺ Runx2⁺ immature osteoblasts accumulated at enthesal sites in AS patients and animal models. At the late stage, increased CaSR⁺ OCN⁺ mature osteoblasts accumulated at enthesal sites (Figure 1H and I, Figure 2G and H, and Figure 3G and H). These results indicate that these CaSR⁺ osteoblasts are potentially involved in the process of pathological new bone formation.

Figure 1H

Figure 2G

Figure 3G

Figure 1I

Figure 2H

Figure 3H

(**Figure 1H and I**) Immunofluorescence analysis of CaSR⁺ (Green) and OCN⁺ (Red) cells in human spinal tissue. (**Figure 2F and G**) Immunofluorescence analysis of CaSR⁺ (Green) and OCN⁺ (Red) cells in PGIS model. (**Figure 3F and G**) Immunofluorescence analysis of CaSR⁺ (Green) and OCN⁺ (Red) cells in DBA/1 model. Data shown as mean \pm SD. ** P < 0.01 compared between groups. Scale bar: 100 μ m.

All these figures have been replaced with more representative image, which have been relabeled as Figure 1H and I, Figure 2G and H, and Figure 3G and H in the revised manuscript.

Reference for this response:

1. Cheng Z, Li A, Tu CL, Maria CS, Szeto N, Herberger A, et al. Calcium-Sensing Receptors in Chondrocytes and Osteoblasts Are Required for Callus Maturation and Fracture Healing in Mice. *J Bone Miner Res.* 2020 Jan; 35(1):143-154.
2. Hannan FM, Kallay E, Chang W, Brandi ML, Thakker RV. The calcium-sensing receptor in physiology and in calcitropic and noncalcitropic diseases. *Nat Rev Endocrinol.* 2018 Dec; 15(1):33-51.

10. Osteophytes are exostoses that form along joint margins, typically intra-articular. How could ectopic cartilage develop to osteophyte (Fig.2D)?

Response:

Thank you for the careful review and we are sorry for this confusion. It is reported that both endochondral ossification and intramembranous ossification are involved in the pathological new bone formation in AS [1, 2]. We intended to show the pathological process of new bone formation at different time points in PGIS model. To avoid confusion, we used the term of “pathological new bone” to replace “osteophyte” to be more precise.

Reference for this response:

1. Sieper J, Poddubny D. Axial spondyloarthritis. *Lancet.* 2017 Jul 1; 390(10089):73-84.
2. Francois RJ, Gardner DL, Degraeve EJ, Bywaters EGL. Histopathologic evidence that sacroiliitis in ankylosing spondylitis is not merely enthesitis - Systematic study of specimens from patients and control subjects. *Arthritis Rheum-US.* 2000 Sep; 43(9):2011-2024.

11. In PGIS mouse model, 24W SOFG staining showed cartilage in the ligament in Fig. 2D and ossification of IVD in Fig. 4B. Confusingly, IVD was intact at 30W in Fig. 2F.

Response:

Thank you for the careful review and we are sorry for this mistake. The representative image of 30 weeks group in Figure 2G was mistakenly used. We have replaced this image with a correct one to demonstrate our results in revised Figure 2G.

Figure 2G

(Figure 2G) H&E, SOFG and Immunofluorescence analyses of the spine specimen of PGIS model at 30 Weeks compare to baseline.

12. It is reasonable to use SMTS model to study unbalanced mechanical force induced enthesopathy. However, SMTS model is not a well-accepted AS model, as AS is not caused by the imperfection of mechanical loading. Moreover, the results of SMTS models reflect enthesopathy other than ectopic bone formation. The manuscript lacks the rationale why SMTS model were used.

Response:

Thank you for the careful review and helpful suggestion. Mechanical loading has been recognized as an important factor that plays a critical role in enthesopathy, which

is the typical pathological feature in AS [1-5]. Previous study has shown that mechanical loading was correlated with both enthesitis and enthesal new bone formation in TNF transgenic mice and collagen antibody-induced arthritis (CAIA) models [6]. However, the primary driver of enthesitis is unclear so far. One hypothesis is that mechanical stress and microdamage may be instrumental in inflammatory enthesitis as a primary driver [7]. To test whether aberrant upregulation of CaSR plays a critical role under this hypothesis, SMTS, a model with enthesopathy and pathological new bone formation due to unbalanced mechanical loading was used [8]. Together with other models used in the current studies, we believe that it could provide more convincing data to support the critical role of CaSR in the pathological process of enthesal new bone formation [9].

The rationale of SMTS model has been added into the discussion section of the revised manuscript (Page 18, Line 19-22; Page 19, Line 1-5).

Reference for this response:

1. Sieper J, Poddubnyy D. Axial spondyloarthritis. *Lancet*. 2017 Jul 1; 390(10089):73-84.
2. Jacques P, McGonagle D. The role of mechanical stress in the pathogenesis of spondyloarthritis and how to combat it. *Best Pract Res Clin Rheumatol*. 2014 Oct; 28(5):703-710.
3. Ramiro S, Landewe R, van Tubergen A, Boonen A, Stolwijk C, Dougados M, et al. Lifestyle factors may modify the effect of disease activity on radiographic progression in patients with ankylosing spondylitis: a longitudinal analysis. *Rmd Open*. 2015 Jan; 1(1).
4. McGonagle D, Stockwin L, Isaacs J, Emery P. An enthesitis based model for the pathogenesis of spondyloarthropathy. additive effects of microbial adjuvant and biomechanical factors at disease sites. *J Rheumatol*. 2001 Oct; 28(10):2155-2159.
5. Aydin SZ, Can M, Alibaz-Oner F, Keser G, Kurum E, Inal V, et al. A relationship between spinal new bone formation in ankylosing spondylitis and the sonographically determined Achilles tendon enthesophytes. *Rheumatol Int*. 2016 Mar; 36(3):397-404.
6. Jacques P, Lambrecht S, Verheugen E, Pauwels E, Kollias G, Armaka M, et al. Proof of concept: enthesitis and new bone formation in spondyloarthritis are driven by mechanical strain and stromal cells. *Annals of the Rheumatic Diseases*. 2014 Feb; 73(2):437-445.
7. Schett G, Lories RJ, D'Agostino MA, Elewaut D, Kirkham B, Soriano ER, et al. Enthesitis: from pathophysiology to treatment. *Nat Rev Rheumatol*. 2017 Nov 21; 13(12):731-741.
8. Wang X, Xie L, Crane J, Zhen G, Li F, Yang P, et al. Aberrant TGF-beta activation in bone tendon insertion induces enthesopathy-like disease. *J Clin Invest*. 2018 Feb 1; 128(2):846-860.

9. Vieira-Sousa E, van Duivenvoorde LM, Fonseca JE, Lories RJ, Baeten DL. Animal Models as a Tool to Dissect Pivotal Pathways Driving Spondyloarthritis. *Arthritis Rheumatol.* 2015 Nov; 67(11):2813-2827.

13. The spontaneously occurring arthritis in DBA/1 mice will develop enthesophyte formation, which is the hallmark of enthesopathy. This model is not suitable for study ectopic bone. How DBA/1 mice are related to AS also needs justification (such as genome wide association studies)

Response:

Thank you for the careful review and constructive comments. Enthesopathy and enthesophyte (pathological new bone) formation are hallmarks of AS [1]. Previous study demonstrated that Aging DBA/1 mice model spontaneously developed arthritis, dactylitis and enthesitis, followed by rigidity of the ankles and enthesophyte formation [2]. This rigidity corresponded histologically to cartilage hyperplasia and subsequent ossification at the bone insertions of the ligamentous components [2, 3]. It indicated that the pathological mechanism of enthesopathy in DBA/1 was similar to that in patients with AS [2, 3]. Moreover, this model has been widely used to investigate various etiology of AS including mechanical stress and adaptive immunity [4, 5].

At present, the diversity of SpA phenotypes sharing peripheral, axial, and extraarticular manifestations (e.g., psoriasis, uveitis, and inflammatory bowel disease) with different degrees of severity precludes the validity of using a single animal model for the study of human SpA [2]. Therefore, we used three animal models to represent different aspects of etiology of AS.

The rationale of DBA/1 model has been added into the discussion section of the revised manuscript (Page 18, Line 14-19).

Reference for this response:

1. Sieper J, Poddubnyy D. Axial spondyloarthritis. *Lancet.* 2017 Jul 1; 390(10089):73-84.
2. Vieira-Sousa E, van Duivenvoorde LM, Fonseca JE, Lories RJ, Baeten DL. Animal Models as a Tool to Dissect Pivotal Pathways Driving Spondyloarthritis. *Arthritis Rheumatol.* 2015

Nov; 67(11):2813-2827.

3. Nordling C, Karlsson-Parra A, Jansson L, Holmdahl R, Klareskog L. Characterization of a spontaneously occurring arthritis in male DBA/1 mice. *Arthritis Rheum.* 1992 Jun; 35(6):717-722.
4. Corthay A, Hansson AS, Holmdahl R. T lymphocytes are not required for the spontaneous development of enthesal ossification leading to marginal ankylosis in the DBA/1 mouse. *Arthritis Rheum.* 2000 Apr; 43(4):844-851.
5. Braem K, Carter S, Lories RJ. Spontaneous arthritis and ankylosis in male DBA/1 mice: further evidence for a role of behavioral factors in "stress-induced arthritis". *Biol Proced Online.* 2012 Dec 19; 14.

14. The baseline described in Fig.4C is very confusing. How treatment group is better than the baseline when there was no ankylosis or ectopic bone.

Response:

Thank you for the careful review and we are sorry for this confusion. We mistakenly used the term of “baseline” in the description of this figure. The description of Fig.4C have been revised to “The incidence of spinal ankylosis and pathological new bone formation were decreased compared to the control group (DMSO administration) at 24 weeks (Figure 4C and D).” (Page 11, Line 1-3)

Figure 4C

Figure 4D

(Figure 4C) Incidence of spinal ankylosis in PGIS model. n=5 per cage of total 3 cages per group. (Figure 4D) Quantitative analysis of structural parameters of new bone by μ CT analysis. Data shown as mean \pm SD. ** P < 0.01, * P < 0.05 compared

15. CaSR antagonist increased the PTH level to decrease OCN+ cells. However, the effect of PTH on bone is anabolic. The authors should explain why the ectopic bone formation were decreased after CaSR antagonist treatment? Were there any

osteoclasts?

Response:

Thank you for the careful review and constructive comments. The purpose of using the CaSR antagonist NPS-2143 in the current study was to observe the effect of CaSR inhibition on the pathological new bone formation in animal models, which was a loss-of-function experiment in vivo. Indeed, as reported by previous study by Gowen M. et al [1], administration of NPS-2143 could increase serum level of PTH. However, they did not observe change of bone mineral density (BMD) in ovariectomized (OVX) model.

In fact, the effect of PTH on bone metabolism is complicated. Both continuous PTH and intermittent PTH increase bone turnover in trabecular and cortical bone, as evidenced by elevations in histomorphometric and biochemical markers of bone resorption and formation [2, 3]. Severe chronic elevations of PTH levels may lead to trabecular bone loss, although continuous PTH treatment often induce a modest increase in cancellous bone [4, 5]. In contrast, intermittent PTH treatment markedly increases trabecular bone volume due to a preponderant stimulation of trabecular bone formation and causes a small loss of cortical bone [6-9]. The biological activities of PTH result from its interaction with various types of cells, including bone marrow stromal cell, osteogenic precursor cell and immune cell [8, 10-12]. PTH directs the osteogenic fate of bone marrow mesenchymal cell through PTH1R signaling [8]. Besides, PTH exerts its anabolic effect on bone partly through orchestrating signaling of local factors including TGF β , Wnts, BMP, and IGF-1 [11, 13-16]. Continuous PTH treatment cause cortical bone loss by enhancing endosteal resorption through stimulation of osteoclast (OC) formation and activity [5, 17]. The catabolic effect of PTH required T cell and is potentiated by inflammatory cytokines [9, 12]. Above all, the exact role of PTH on bone metabolism is so far still under investigation and beyond the goal of current study.

The role of PTH in pathological new bone formation in AS is also unclear. Whether there is difference of serum PTH level in AS patients compared to healthy

control remains controversial, indicating a complicated relationship between PTH and pathological new bone formation in AS [18-21]. To determine the effect of PTH on pathological new bone formation in AS, our group have been conducting another targeted study. In DBA/1 model intermitted administration of PTH (1-34) (iPTH) did not increase pathological new bone volume (Figure R2). Similarly, in SMTS model, we found that iPTH had no effect on pathological new bone formation, as both areas of UF and CF were not increased after iPTH administration compared to vehicle group (Figure R3) (unpublished data).

Figure R2

Figure for reviewers removed

Figure R2. (A) μ CT analysis of DBA/1 model. (B) Quantification of new bone volume in DBA/1 model. n=6 per group. Data shown as mean \pm SD. NS: $p \geq 0.05$ compared between groups.

Figure R3

Figure for reviewers removed

Figure R2. (A) μ CT analysis of SMTS model. (B) SOFG staining of SMTS model. (C) Quantification of CF (calcified fibrocartilage) and UF (uncalcified fibrocartilage). n=5. Data shown as mean \pm SD. ** $P < 0.01$ compared between groups. Scale bar: 100 μ m.

In addition, a PTH conditional knockout (*CAGGcreERTM;PTH^{fl/fl}*) mice model was established. Tamoxifen was administered daily 1 week before tenotomy to induce knockdown of PTH in SMTS model and continually for 8 weeks. The results showed that ablation of PTH did not reduce the pathological new bone formation in *PTH CKO* SMTS model. In contrast, administration of NPS-2143 suppressed the pathological new bone formation in *PTH CKO* SMTS model, as confirmed by reduced enthesophyte formation and decreased UF and CF zones (Figure R4A to C).

Therefore, these results strongly supported that NPS-2143 suppressed pathological new bone formation in a PTH independent manner. Since investigation

the effect of iPTH to pathological new bone formation is an ongoing study, we would rather not to present the relative results in the current study.

Figure R4

Figure R4. (A) μ CT images of the PCT in *CAGGcreERTM;PTH^{fl/fl}* SMTS model, n=5 per group. (B) SOFG staining of Achilles tendon enthesis compartment in *CAGGcreERTM;PTH^{fl/fl}* SMTS model. (C) Quantitative analysis of area of UF and CF. Data shown as mean \pm SD. ** P < 0.01 compared between groups. Scale bar: 100 μ m.

Previous studies reported that activation of CaSR in osteoclasts affect its bone resorption ability differently depending on different biological condition. High concentration of calcium inhibited activated CaSR to inhibit osteoclastogenesis and induce apoptosis of osteoclast while lower dose of calcium activated CaSR to promoted osteoclastic differentiation [10, 22, 23]. As suggested, we further performed immunofluorescence staining of DBA/1 model to determine the existence of osteoclasts. The results showed that CaSR was expressed in CTSK⁺ osteoclast at bone erosion area in DBA/1 model. Nonetheless, we fail to observe upregulation of CaSR

in CTSK⁺ osteoclasts (Figure R5). Furthermore, it is reported that inhibition of osteoclasts does not prevent joint ankylosis due to pathological new bone formation in DBA/1 model [24]. Collectively, these results indicate that inhibitory effect of CaSR antagonist on pathological new bone formation is not likely involve osteoclastogenesis.

Figure R5

Figure R5. Immunofluorescence analysis of CaSR⁺ and CTSK⁺ cells at bone erosion site in DBA/1 model. Data shown as mean \pm SD. NS: not significant, $p \geq 0.05$.

Reference

for this

response:

1. Gowen M, Stroup GB, Dodds RA, James IE, Votta BJ, Smith BR, et al. Antagonizing the parathyroid calcium receptor stimulates parathyroid hormone secretion and bone formation in osteopenic rats. *J Clin Invest.* 2000 Jun; 105(11):1595-1604.
2. Dempster DW, Cosman F, Parisien M, Shen V, Lindsay R. Anabolic actions of parathyroid hormone on bone. *Endocr Rev.* 1993 Dec; 14(6):690-709.
3. Iida-Klein A, Lu SS, Kapadia R, Burkhart M, Moreno A, Dempster DW, et al. Short-term continuous infusion of human parathyroid hormone 1-34 fragment is catabolic with decreased trabecular connectivity density accompanied by hypercalcemia in C57BL/J6 mice. *Journal of Endocrinology.* 2005 Sep; 186(3):549-557.
4. Zhou H, Shen V, Dempster DW, Lindsay R. Continuous parathyroid hormone and estrogen

- administration increases vertebral cancellous bone volume and cortical width in the estrogen-deficient rat. *J Bone Miner Res.* 2001 Jul; 16(7):1300-1307.
5. Lotinun S, Evans GL, Bronk JT, Bolander ME, Wronski TJ, Ritman EL, et al. Continuous parathyroid hormone induces cortical porosity in the rat: effects on bone turnover and mechanical properties. *J Bone Miner Res.* 2004 Jul; 19(7):1165-1171.
 6. Neer RM, Arnaud CD, Zanchetta JR, Prince R, Gaich GA, Reginster JY, et al. Effect of parathyroid hormone (1-34) on fractures and bone mineral density in postmenopausal women with osteoporosis. *N Engl J Med.* 2001 May 10; 344(19):1434-1441.
 7. Leong I. Effects of PTH on bone composition. *Nat Rev Endocrinol.* 2018 Dec; 14(12):689.
 8. Fan Y, Hanai JI, Le PT, Bi R, Maridas D, DeMambro V, et al. Parathyroid Hormone Directs Bone Marrow Mesenchymal Cell Fate. *Cell Metab.* 2017 Mar 7; 25(3):661-672.
 9. Li JY, D'Amelio P, Robinson J, Walker LD, Vaccaro C, Luo T, et al. IL-17A Is Increased in Humans with Primary Hyperparathyroidism and Mediates PTH-Induced Bone Loss in Mice. *Cell Metab.* 2015 Nov 3; 22(5):799-810.
 10. Shu L, Ji J, Zhu Q, Cao G, Karaplis A, Pollak MR, et al. The calcium-sensing receptor mediates bone turnover induced by dietary calcium and parathyroid hormone in neonates. *J Bone Miner Res.* 2011 May; 26(5):1057-1071.
 11. Yu B, Zhao X, Yang C, Crane J, Xian L, Lu W, et al. Parathyroid hormone induces differentiation of mesenchymal stromal/stem cells by enhancing bone morphogenetic protein signaling. *J Bone Miner Res.* 2012 Sep; 27(9):2001-2014.
 12. Gao Y, Wu X, Terauchi M, Li JY, Grassi F, Galley S, et al. T cells potentiate PTH-induced cortical bone loss through CD40L signaling. *Cell Metabolism.* 2008 Aug 6; 8(2):132-145.
 13. Wu X, Pang L, Lei W, Lu W, Li J, Li Z, et al. Inhibition of Sca-1-positive skeletal stem cell recruitment by alendronate blunts the anabolic effects of parathyroid hormone on bone remodeling. *Cell Stem Cell.* 2010 Nov 5; 7(5):571-580.
 14. Guo J, Liu M, Yang D, Bouxsein ML, Saito H, Galvin RJ, et al. Suppression of Wnt signaling by Dkk1 attenuates PTH-mediated stromal cell response and new bone formation. *Cell Metab.* 2010 Feb 3; 11(2):161-171.
 15. Qiu T, Wu X, Zhang F, Clemens TL, Wan M, Cao X. TGF-beta type II receptor phosphorylates PTH receptor to integrate bone remodelling signalling. *Nat Cell Biol.* 2010 Mar; 12(3):224-234.
 16. Bikle DD, Sakata T, Leary C, Elalieh H, Ginzinger D, Rosen CJ, et al. Insulin-like growth factor I is required for the anabolic actions of parathyroid hormone on mouse bone. *Journal of Bone and Mineral Research.* 2002 Sep; 17(9):1570-1578.
 17. Bilezikian JP, Bandeira L, Khan A, Cusano NE. Hyperparathyroidism. *Lancet.* 2018 Jan 13; 391(10116):168-178.
 18. Orsolini G, Adami G, Rossini M, Ghellere F, Caimmi C, Fassio A, et al. Parathyroid hormone is a determinant of serum Dickkopf-1 levels in ankylosing spondylitis. *Clin Rheumatol.* 2018 Nov; 37(11):3093-3098.
 19. Lange U, Jung O, Teichmann J, Neeck G. Relationship between disease activity and serum levels of vitamin D metabolites and parathyroid hormone in ankylosing spondylitis. *Osteoporosis Int.* 2001; 12(12):1031-1035.

20. Gonullu E, Bilge NSY, Cansu DU, Bekmez M, Musmul A, Akcar N, et al. Risk factors for urolithiasis in patients with ankylosing spondylitis: a prospective case-control study. *Urolithiasis*. 2017 Aug; 45(4):353-357.
21. Baskan BM, Dogan YP, Sivas F, Bodur H, Ozoran K. The relation between osteoporosis and vitamin D levels and disease activity in ankylosing spondylitis. *Rheumatol Int*. 2010 Jan; 30(3):375-381.
22. Mentaverri R, Yano S, Chattopadhyay N, Petit L, Kifor O, Kamel S, et al. The calcium sensing receptor is directly involved in both osteoclast differentiation and apoptosis. *Faseb Journal*. 2006 Dec; 20(14):2562-+.
23. Goltzman D, Hendy GN. The calcium-sensing receptor in bone--mechanistic and therapeutic insights. *Nat Rev Endocrinol*. 2015 May; 11(5):298-307.
24. Lories RJ, Derese I, Luyten FP. Inhibition of osteoclasts does not prevent joint ankylosis in a mouse model of spondyloarthritis. *Rheumatology (Oxford)*. 2008 May; 47(5):605-608.

16. As the CaSR antagonist could regulate PTH level. Are there any side effect of CaSR antagonist?

Response:

In the current study, NPS-2143 was used as an experimental tool to investigate the role of CaSR in pathological new bone formation. We did not intend to determine the translational potential of this specific chemical inhibitor. However, previous studies indicate that side effects of CaSR antagonist include depression of air flow resistance, increasing mean arterial pressure and inhibition of the secretion of calcitonin, which might be due to the change of serum level of PTH [1-3]. Thus, a more specific CaSR antagonist targeting to osteo-lineage cells should be designed to better meet the need for medical translation.

Reference for this response:

1. Loupy A, Ramakrishnan SK, Wootla B, Chambrey R, de la Faille R, Bourgeois S, et al. PTH-independent regulation of blood calcium concentration by the calcium-sensing receptor. *J Clin Invest*. 2012 Sep; 122(9):3355-3367.
2. Rybczynska A, Lehmann A, Jurska-Jasko A, Boblewski K, Orlewska C, Foks H, et al. Hypertensive effect of calcilytic NPS 2143 administration in rats. *J Endocrinol*. 2006 Oct; 191(1):189-195.
3. Poon SF, St Jean DJ, Harrington PE, Henley C, Davis J, Morony S, et al. Discovery and Optimization of Substituted 1-(1-Phenyl-1H-pyrazol-3-yl)methanamines as Potent and

17. The English is largely inadequate, and thus the manuscript needs significant editing.

Response:

Thank you for the careful review and helpful suggestion. The language in the revised manuscript has been polished by native English-speaking professionals from AJE company. We hope that the language is more accurate and eliminates any misunderstandings.

Reviewer 2

1. Given that enthesal tissues give rise to bone formation through the process of endochondral ossification, osteo-chondro progenitor cells first differentiate into chondrocytes and subsequently differentiate into osteoblast-making bone matrix. The authors also described the importance of endochondral ossification as a pathological process of ectopic bone formation in the Discussion. However, the authors mainly focus on osteoblast differentiation. As chondrocytes also highly express CaSR (Fig 1-3), it would be interesting to see the effect of CaSR on chondrocytes differentiation from progenitor cells. If CaSR induces the differentiation into chondrocytes, it would suggest that CaSR is involved in the whole process of endochondral ossification. Otherwise bone formation may occur through intramembrane ossification rather than endochondral ossification.

Response:

Thank you for the careful review and kind suggestion. Previous studies showed that CaSR plays a critical role in modulating chondrogenic differentiation and endochondral ossification [1, 2]. Knock down of CaSR in mesenchymal stem cells (MSCs) inhibited its chondrogenesis, while knock down of CaSR in chondrocytes impacted its terminal differentiation to hypertrophic chondrocytes and secretion of bone-forming matrix [1]. However, hyperstimulation of CaSR did not enhance

chondrogenic differentiation of MSCs. These findings indicate that CaSR is essential but not sufficient for endochondral ossification [3].

Prompted by the respected reviewer, we performed additional experiments. Consistent with previous studies, CaSR antagonist NPS-2143 suppressed chondrogenic differentiation of ATDC5 cell (chondrogenic cell line). However, CaSR agonist SR failed to enhance chondrogenic differentiation in chondrogenic precursor cells (Figure EV2A and B). In addition, immune cytokines could upregulate expression of CaSR in osteoblasts but failed to increase the expression of CaSR in chondrocytes (Figure EV2C). These results suggested that the interaction between inflammation and upregulation of CaSR mainly occurred in osteoblasts rather than chondrocytes during the process of pathological new bone formation.

Figure EV2. The role of CaSR⁺ chondrocytes in AS models. **(A)** Alcian Blue staining of ATDC5 cells. **(B)** RT-qPCR analysis of chondrogenic markers in ATDC5 cells. **(C)** RT-qPCR analysis of inflammatory cytokines in ATDC5 cells. Data shown as mean \pm SD. ** $P < 0.01$ compared between groups, NS: not significant, $p \geq 0.05$ compared between groups.

Inspired by the reviewer, we recognized that the suppressive effect of CaSR inhibition on pathological new bone formation involves two different mechanisms. On one hand, inhibition of CaSR suppressed chondrogenic differentiation of the precursor cells, terminal hypertrophy of the chondrocytes and secretion of bone-forming matrix, which negatively impacted the the process of endochondral ossification. On the other hand, inhibition of CaSR directly suppressed intramembrane

ossification. Therefore, it further indicated that CaSR might serve as a therapeutic target for pathological new bone formation with strong potential.

Additional results have been added into revised manuscript (Page 12, Line 2-12; Page 14, Line20-22; Page 15, Line 1; Page 20, Line 9-22; Page 21, Line 1-5)

Reference for this response:

1. Cheng Z, Li A, Tu CL, Maria CS, Szeto N, Herberger A, et al. Calcium-Sensing Receptors in Chondrocytes and Osteoblasts Are Required for Callus Maturation and Fracture Healing in Mice. *J Bone Miner Res.* 2020 Jan; 35(1):143-154.
2. Goltzman D, Hendy GN. The calcium-sensing receptor in bone--mechanistic and therapeutic insights. *Nat Rev Endocrinol.* 2015 May; 11(5):298-307.
3. Sarem M, Heizmann M, Barbero A, Martin I, Shastri VP. Hyperstimulation of CaSR in human MSCs by biomimetic apatite inhibits endochondral ossification via temporal down-regulation of PTH1R. *Proc Natl Acad Sci U S A.* 2018 Jul 3; 115(27):E6135-E6144.

2. Figure 1-3: Although AS patients and AS animal models have a significant population of CaSR-positive cells, CaSR is physiologically wide-expressed in various tissues to regulate important cellular functions such as maintaining systemic calcium homeostasis. The difference in the positive ratio between AS and healthy controls may be attributed to the difference in the total cell number of the tissues, rather than high positive ratio. Instead of providing the number of positive cells, percentages of positive cells would be ideal to show the upregulated CaSR. Also, providing a negative control would be helpful. Furthermore, as enthesitis lesions have significant immune cell infiltration, it would be informative to characterize the immune cell population with representative markers (such as CD3, CD19, and CD68) to distinguish the osteo-progenitor cells from immune cells in the histology slides in IHC or IF.

Response:

Thank you for the careful review and constructive suggestion. We have added percentages of CaSR positive cells to show the aberrant upregulated CaSR at new

bone-forming sites in revised Figure 1G and I, Figure 2F and H, Figure 3F and H; EV1G and I.

(**Figure 1G and H**). Quantitative analysis CaSR⁺ cell percentage in human spinal tissue. (**Figure 2F and H**). Quantitative analysis CaSR⁺ cell percentage in PGIS model. (**Figure 3F and H**). Quantitative analysis CaSR⁺ cell percentage in DBA/1 model. (**Figure EV1G and I**). Quantitative analysis CaSR⁺ cell percentage in SMTS model.

To better demonstrated the upregulation of CaSR, human spinal tissues were collected to detected the expression of CaSR by RT-PCR and western blot analysis. The results showed that expression of CaSR was upregulated in spinal tissues from AS patients compared to control group (Figure 1D). Similarly, upregulation of CaSR was observed in PGIS, DBA/1 and SMTS models, as confirmed by PCR analysis (Figure 2D and 3D). Taken together, these results suggest that CaSR is upregulated in AS patients and animal models. (Page 7, Line 9-10; Page 8, Line 7-8; Page 9, Line 5-6; Page 10, Line 2-3).

(**Figure 1D**). RT-qPCR analysis of expression of CaSR in human spinal tissues. (**Figure 2D**). RT-qPCR analysis of expression of CaSR in PGIS model. (**Figure 3D**). RT-qPCR analysis of expression of CaSR in DBA/1 model. (**Figure EV1E**) RT-qPCR analysis of expression of CaSR in SMTS model. Data shown as mean \pm SD. ** $P < 0.01$ compared between groups.

As suggested, IgG2 was used as negative control in IF. Negative control staining of CaSR in PGIS, DBA/1 and SMTS models are presented below (Figure R6).

Figure R6

Figure R6. Negative control staining of CaSR in animal models. (**A**) Staining is absent with omission of the primary antibody in PGIS model. (**B**) Staining is absent with omission of the primary antibody in DBA/1 model. (**C**) Staining is absent with omission of the primary antibody in SMTS model.

Immunofluorescence staining of CD45 (hemopoietic cells) and Runx2 (osteogenic progenitor cells) was performed to better distinguish the osteo-progenitor cells from immune cells in Figure EV4A to C. The results showed that CD45⁺ cells accumulated around Runx2⁺ cells during new bone formation. The proximity of CD45⁺ cells and Runx2⁺ cells provided the opportunity for communication between these two types of cells. The relative information has been added into the result section of revised manuscript (Page 16, Line 15-19; Page 22, Line 7-10) and Extended view data.

Figure EV4A to C

Figure EV4. (A) Immunofluorescence analyses of CD45⁺ and Runx2⁺ cells at spinal enthesal site in human spinal tissues. (B) Immunofluorescence analyses of CD45⁺ and Runx2⁺ cells at spinal enthesal site in PGIS model. (C) Immunofluorescence analyses of CD45⁺ and Runx2⁺ cells at ankle enthesal site in DBA/1 model. Data shown as mean ± SD. ** P < 0.01

3. Fig2B: There is a difference in the percentage of spinal ankylosis depending on cages. Does the degree of inflammation make the difference? Or could this reflect different microbiota? I would suggest providing the correlation between the severity of inflammation and percentage of bone formation in the spine.

Response:

Thank you for the careful review and kind suggestion. According to previous studies, the incidence of spinal ankylosis is unstable, varies from 64% to 100% at 24 weeks [1, 2]. The incidence of spinal ankylosis varies between cages (20 cages at all) from to 65% to 85% at 24 weeks in current and previous studies of our group. According to statistical analysis, the various incidence of spinal ankylosis between cages did not affect the significance of difference between groups.

The relationship between inflammation and new bone formation in AS is still an enigma [3]. Previous study showed that inflammation is decreased along with excessive tissue and ectopic chondrocyte formation driven by chondroid ossification in PGIS model, indicating that inflammation is not positive correlated with new bone formation [2]. In AS patients, successful treatment of signs and symptoms of AS with subsequent improvement in physical function and quality of life-using TNF blockers has dramatically changed the clinical impact of disease in patients with AS [4]. However, TNFi application during radiographic interval in AS has minimal effect on spinal radiographic progression despite its clinically relevant inhibitory effect on spinal radiographic changes if applied before radiographic interval. Several animal experiments and clinical studies have found that inflammation and new bone formation may be two independent processes in the natural course of ankylosis [5, 6]. New bone formation still existed after resolution of inflammation in patients with AS [7]. Therefore, there is uncoupling of inflammation and spinal ankylosis to some extent.

Another interesting topic of the relationship between microbiota and new bone formation is beyond the purpose of our study. So far, there is no valid conclusion for the role of microbiota and inflammation in the development of new bone formation in AS [8-11]. Thus, further comprehensive study focusing on this topic is needed.

Although we further conducted additional analysis of the correlation between inflammation severity and the incidence of spinal ankylosis, the result could not lead to the conclusion that they were positively correlated due to the rule of the histological score system (Figure R7). According to this scoring system (score 1, enthesitis, inflammatory cell accumulation around the IVD and/or infiltration of the annulus fibrosus; score 2, < 50% absorption/erosion of the IVD; score 3, essentially complete resorption (> 50%) of the IVD; score 4, cartilaginous/bony ankylosis), degree of inflammation and bony ankylosis was evaluated, meaning that the high score is based on new bone formation [12].

Figure R7

Figure R7. Correlation analysis between histological score and incidence of spinal ankylosis.

As suggested, to determine the correlation between the severity of inflammation and percentage of bone formation, we additionally evaluated degree of inflammation with another inflammation scoring system, which only capture the inflammatory changes that occur within the vertebral joints of the spine in this AS model (score 0, Normal; score 1, Minor infiltration of inflammatory cells at periphery of the joint;

score 2, Moderate infiltration – inflammatory pannus < 50 % joint area; score 3, Marked infiltration – inflammatory pannus > 50 % joint area) [2]. The result showed that average inflammation score raised to 1.47 ± 0.77 at 16 weeks and reduced to 0.57 ± 0.50 at 24 weeks (Figure R8). Correlation analysis showed that there was no correlation between inflammation score and incidence of spinal ankylosis (Figure R9). Therefore, we could not draw the conclusion that inflammation is correlated with incidence of spinal ankylosis.

Figure R8

Figure R8. Inflammation scores of PGIS model. Data shown as mean \pm SD. NS: not significant, $p \geq 0.05$.

Figure R9

Figure R9. Correlation analysis between inflammation score and incidence of spinal ankylosis.

Referecne for this response:

1. Bardos T, Szabo Z, Czipri M, Vermes C, Tunyogi-Csapo M, Urban RM, et al. A longitudinal study on an autoimmune murine model of ankylosing spondylitis. *Ann Rheum Dis*. 2005 Jul; 64(7):981-987.
2. Tseng HW, Pitt ME, Glant TT, McRae AF, Kenna TJ, Brown MA, et al. Inflammation-driven bone formation in a mouse model of ankylosing spondylitis: sequential not parallel processes. *Arthritis Res Ther*. 2016 Jan 29; 18:35.
3. Lories RJ, Dougados M. Inflammation and ankylosis: still an enigmatic relationship in spondyloarthritis. *Ann Rheum Dis*. 2012 Mar; 71(3):317-318.
4. Sieper J, Poddubnyy D. Axial spondyloarthritis. *Lancet*. 2017 Jul 1; 390(10089):73-84.
5. Poddubnyy D, Sieper J. Mechanism of New Bone Formation in Axial Spondyloarthritis. *Curr Rheumatol Rep*. 2017 Sep; 19(9):55.
6. Lories RJ, Derese I, de Bari C, Luyten FP. Evidence for uncoupling of inflammation and joint remodeling in a mouse model of spondylarthritis. *Arthritis Rheum*. 2007 Feb; 56(2):489-497.
7. Magrey MN, Khan MA. The Paradox of Bone Formation and Bone Loss in Ankylosing Spondylitis: Evolving New Concepts of Bone Formation and Future Trends in Management. *Curr Rheumatol Rep*. 2017 Apr; 19(4):17.
8. Ranganathan V, Gracey E, Brown MA, Inman RD, Haroon N. Pathogenesis of ankylosing spondylitis - recent advances and future directions. *Nat Rev Rheumatol*. 2017 Jun; 13(6):359-367.
9. Cuthbert RJ, Watad A, Fragkakis EM, Dunsmuir R, Loughenbury P, Khan A, et al. Evidence that tissue resident human enthesis gamma delta T-cells can produce IL-17A independently of IL-23R transcript expression. *Annals of the Rheumatic Diseases*. 2019 Nov; 78(11):1559-1565.
10. Gravallesse EM, Schett G. Effects of the IL-23-IL-17 pathway on bone in spondyloarthritis. *Nat Rev Rheumatol*. 2018 Nov; 14(11):631-640.
11. Li X, Wang JR, Zhan ZP, Li SB, Zheng ZM, Wang TP, et al. Inflammation Intensity-Dependent Expression of Osteoinductive Wnt Proteins Is Critical for Ectopic New Bone Formation in Ankylosing Spondylitis. *Arthritis Rheumatol*. 2018 Jul; 70(7):1056-1070.
12. Bardos T, Szabo Z, Czipri M, Vermes C, Tunyogi-Csapo M, Urban RM, et al. A longitudinal study on an autoimmune murine model of ankylosing spondylitis. *Ann Rheum Dis*. 2005 Jul; 64(7):981-987.
13. Tseng HW, Pitt ME, Glant TT, McRae AF, Kenna TJ, Brown MA, et al. Inflammation-driven bone formation in a mouse model of ankylosing spondylitis: sequential not parallel processes. *Arthritis Res Ther*. 2016 Jan 29; 18:35.

4. Fig3 A, B: I would suggest using the term "enthesophytes" instead of "osteophytes". The term of "osteophytes" gives readers the impression of osteoarthritis rather than spondylarthritis. Based on Fig 3B and 3D, it seems that all bone formation is initiated from the areas of enthesal tissues. Therefore, enthesophytes would be a precise term

to describe the ectopic bone formation.

Response:

Thank you for the careful review and kind suggestion. We used the term of “ectopic bone formation” is based on the understanding that “ectopic bone” is ossification of tissues outside their usual origins in soft tissues such as muscle, subcutaneous tissue, and fibrous tissue adjacent to joints [1]. However, it indeed caused confusion under the reminder from this respected reviewer and reviewer 1. We again go through the existing literatures on this research field and found that previous studies preferred to use the term of “new bone formation” in PGIS model and “enthesophyte” in DBA/1 and SMTS models [2-8]. Since multiple animal models with different types of hypothetical pathogenesis of AS were used in the current study, we intend to replaced “ectopic bone” and “osteophyte” with “pathological new bone” to generally describe the new bone formation in these animal models.

References for this response

1. Scott MA, Levi B, Askarinam A, Nguyen A, Rackohn T, Ting K, et al. Brief review of models of ectopic bone formation. *Stem Cells Dev.* 2012 Mar 20; 21(5):655-667.
2. van der Heijde D, Machado P, Braun J, Hermann KGA, Baraliakos X, Hsu B, et al. MRI inflammation at the vertebral unit only marginally predicts new syndesmophyte formation: a multilevel analysis in patients with ankylosing spondylitis. *Annals of the Rheumatic Diseases.* 2012 Mar; 71(3):369-373.
3. Sieper J, Poddubnyy D. Axial spondyloarthritis. *Lancet.* 2017 Jul 1; 390(10089):73-84.
4. Benjamin M, McGonagle D. The anatomical basis for disease localisation in seronegative spondyloarthropathy at entheses and related sites. *J Anat.* 2001 Nov; 199(Pt 5):503-526.
5. Benjamin M, McGonagle D. Histopathologic changes at "synovio-enthesal complexes" suggesting a novel mechanism for synovitis in osteoarthritis and spondylarthritis. *Arthritis Rheum-U.S.* 2007 Nov; 56(11):3601-3609.
6. Glant TT, Radacs M, Nagyri G, Olasz K, Laszlo A, Boldizsar F, et al. Proteoglycan-induced arthritis and recombinant human proteoglycan aggrecan G1 domain-induced arthritis in BALB/c mice resembling two subtypes of rheumatoid arthritis. *Arthritis Rheum.* 2011 May; 63(5):1312-1321.
7. Bardos T, Szabo Z, Czipri M, Vermes C, Tunyogi-Csapo M, Urban RM, et al. A longitudinal study on an autoimmune murine model of ankylosing spondylitis. *Ann Rheum Dis.* 2005 Jul; 64(7):981-987.
8. Lories RJ. Animal models of spondyloarthritis. *Curr Opin Rheumatol.* 2006 Jul; 18(4):342-346.

5. Fig3. Again, it would be ideal to provide the correlation between the severity of inflammation and bone formation.

Response:

Thank you for the careful review and kind suggestion. The clinical severity score is not correlated with incidence of ankle enthesophyte in DBA/1 model, as showed below (Figure R10). It is previously reported that severity of inflammation is not correlated with new bone formation in AS [1, 2]. Therefore, the relationship between inflammation and new bone formation is still an enigma.

Figure R10

Figure R10. Correlation analysis of Clinical severity score and incidence of ankle enthesophyte.

Reference for this response:

1. Sieper J, Poddubnyy D. Axial spondyloarthritis. *Lancet*. 2017 Jul 1; 390(10089):73-84.
2. Lories RJ, Dougados M. Inflammation and ankylosis: still an enigmatic relationship in spondyloarthritis. *Ann Rheum Dis*. 2012 Mar; 71(3):317-318.

6. FigEV2: Given the inflammation severity is decreased in the group treated with NPS-2143 (CaSR antagonist) compared to controls, NPS-2143 seems to have a potential to suppress inflammation in the tissue. This suggests that less percentages of bone formation may be due to less inflammation compared to control. Fig 6 also

shows that various inflammatory cytokines increase the expression of CaSR. I would suggest performing an experiment to test the direct effect of CaSR on bone formation or osteocyte differentiation under non-inflammatory conditions. A potential experiment would be to compare bone formation markers (RUNX2/OCN) between CaSR-knockout and WT cells with or without inflammatory condition. Alternatively, the authors could check the bone formation markers in CaSR-overexpressed MC3T3-E1 cells treated with stat3 and pp65 siRNAs.

Response:

Thank you for the careful review and kind suggestion. As mentioned above in response to question 3, inflammation is not positively correlated with new bone formation. Besides, histological score system evaluates the histological severity, including bone erosion and cartilaginous/bony ankylosis (histological score system: score 1, enthesitis, inflammatory cell accumulation around the IVD and/or infiltration of the annulus fibrosus; score 2, < 50% absorption/erosion of the IVD; score 3, essentially complete resorption (> 50%) of the IVD; score 4, cartilaginous/bony ankylosis) [1]. Therefore, the reduced histological score by NPS-2143 was mainly attributed to the reduction of new bone formation rather than inflammation.

It is well-acknowledged that activation of CaSR drive osteogenic differentiation in under non-inflammatory conditions [2]. As suggested, we perform alizarin red staining and RT-qPCR analysis to compare bone formation makers (Runx2, Osx, ALP and OCN) between CaSR-knockout and wild type MC3T3-E1 cells with or without inflammatory stimulation. The result showed that the osteogenic potential of MC3T3-E1 cells was suppressed by siCaSR treatment with or without TNF α stimuli (10ng/ml), as proved by reduced calcium deposit and expression of osteogenic markers in both groups (Figure EV4D and E). These results indicate that knockdown of CaSR attenuates osteogenic potential of osteogenic precursor cells under inflammatory or non-inflammatory conditions. Besides, overexpression of CaSR in MC3T3-E1 cells promote its osteogenic potential compared to neagative control.

Transfection with p65 or Stat3 siRNA did not affect the osteogenic effect of CaSR overexpression (Figure EV4F). These results indicate that the osteogenic effect of CaSR overexpression is independent of p65 or Stat3 pathways.

The relative information has been added into the result section of revised manuscript and Figure EV4. (Page 15, Line 2-6, Line 16-20; Page 19, Line 16-18)

Figure EV4D

Figure EV4E

Figure EV4. (D) Quantification of Alizarin Red staining and RT-qPCR analysis of osteogenesis markers of MC3T3-E1 cells. **(E)** Quantification of Alizarin Red staining and RT-qPCR analysis of osteogenesis markers of MC3T3-E1 cells. Data shown as mean ± SD. ** P < 0.01 compared between groups.

Figure EV4F

Figure EV4F. Quantification of Alizarin Red staining and RT-qPCR analysis of osteogenesis markers CaSR-overexpressed MC3T3-E1 cells treated with Stat3 and p65 siRNAs. EV: empty vector. NC: negative control. Data shown as mean \pm SD. ** $P < 0.01$ compared between groups.

Reference for this response:

1. Bardos T, Szabo Z, Czipri M, Vermes C, Tunyogi-Csapo M, Urban RM, et al. A longitudinal study on an autoimmune murine model of ankylosing spondylitis. *Ann Rheum Dis.* 2005 Jul; 64(7):981-987.
2. Goltzman D, Hendy GN. The calcium-sensing receptor in bone--mechanistic and therapeutic insights. *Nat Rev Endocrinol.* 2015 May; 11(5):298-307.

7. Fig 6: Based on the figure, it seems that general inflammation increases the expression of CaSR in osteoblasts through NF- κ B and JAK/STAT3 signaling pathways regardless of the type of cytokine. The authors described that these cytokines are "AS-related inflammatory cytokines", which could be challenged in terms of sensitivity and specificity. There was no screening test performed to detect the cytokines. In addition, those cytokines are also strongly related to other forms of inflammatory arthritis such as rheumatoid arthritis. Although NF- κ B and JAK/STAT3

signaling are indispensable pathways in RA, RA has bone erosive changes unlike AS and lacks osteoproliferation. It is possible that CaSR is upregulated not only osteoblasts but osteoclasts or any other cells. I would suggest testing the expression of CaSR in bone erosion areas to see the activation of osteoclasts or immune cells. It is very important point to see the specific role of CaSR in AS. As DBA/1 arthritis model is also used for the model of RA, the authors will be able to address this question.

Response:

Thank you for the careful review and kind suggestion. We have replaced the "AS-related inflammatory cytokines" with "inflammatory cytokines " to avoid confusion and misunderstanding [1-4]. (Page 14, Line 10-13)

Previous studies indicate that multiple inflammatory cytokines, including TNF α , IL-1 β , IL-17A, IL-22 and IL-23 are involved in the pathological process of new bone formation in AS [1, 5-10]. As suggested, tissue samples were collected to perform screening test to detect these inflammatory cytokines. RT-qPCR analysis showed that the expression of these cytokines at mRNA levels were increased in enthesal tissues from PGIS, DBA/1 and SMTS models (Figure EV4G to I). These results indicated that these inflammatory cytokines might contribute to the upregulation of CaSR in osteoblasts. (Page 14, Line 14-15)

Figure EV4. (G) RT-qPCR analysis of inflammatory cytokines in PGIS model. (H) RT-qPCR analysis inflammatory cytokines in DBA/1 model. (I) RT-qPCR analysis of inflammatory cytokines in SMTS model.

The role of CaSR in osteoclastogenesis and osteoclast function is complicated. Although CaSR was also found to be expressed in osteoclast precursor cells, pre-osteoclasts, and osteoclasts *in vitro*, only a minority of osteoclasts were found expressing CaSR *in vivo* [11-13]. Activation of CaSR in osteoclasts affect its bone resorption ability differently depending on different biological condition. High concentration of calcium activated CaSR to inhibit osteoclastogenesis and induce apoptosis of osteoclast, while lower dose of calcium activated CaSR to promoted osteoclastic differentiation [14-16].

The role of osteoclasts in AS is still unclear. Some studies demonstrated that osteoclastogenesis was suppressed in AS patients, while other studies revealed that osteoclastogenesis was significantly increased at bone erosion sites [17, 18]. However, importantly, it is reported that inhibition of osteoclasts does not prevent joint ankylosis due to pathological new bone formation in DBA/1 model [19].

As suggested, we further detected the existence of osteoclasts and the expression of CaSR in osteoclasts in the bone erosion sites in DBA/1 model. The results showed that CaSR was expressed in CTSK⁺ osteoclast at bone erosion area in DBA/1 model. Nonetheless, we fail to observe upregulation of CaSR in CTSK⁺ osteoclasts at different stages of the pathological process (Figure R5). Similarly, we further detected the expression of CaSR in CD45⁺ immune cells at enthesal sites of hind paws in DBA/1 model. The results showed that CaSR was expressed in CD45⁺ immune cells. Nonetheless, we fail to observe upregulation of CaSR in CD45⁺ immune cells (Figure R11). In addition, multiple immune cytokines including TNF α , IL-1 β , IL-17A, IL-22 and IL-23 failed to significantly increase the expression of CaSR in human peripheral blood mononuclear cell (PBMC) (Figure R12). Taken together, these results indicate that upregulation of CaSR mainly happened in osteoblasts instead of osteoclasts or other immune cells in the process of pathological new bone formation.

Figure R5

Figure R5. Immunofluorescent analysis of CaSR⁺ and CTSK⁺ cells at bone erosion site in DBA/1 model. Data shown as mean \pm SD. NS: not significant, $p \geq 0.05$.

Figure R11

Figure R11. Immunofluorescent analysis of CaSR⁺ and CD45⁺ cells in DBA/1 model. Data shown as mean \pm SD. NS: not significant, $p \geq 0.05$.

Figure R12

Figure R12. RT-qPCR analysis of CaSR expression in human PBMC with cytokines stimulation. Data shown as mean \pm SD. NS: not significant, $p \geq 0.05$.

Reference for this response:

1. Sieper J, Poddubny D. Axial spondyloarthritis. *Lancet*. 2017 Jul 1; 390(10089):73-84.
2. Ranganathan V, Gracey E, Brown MA, Inman RD, Haroon N. Pathogenesis of ankylosing spondylitis - recent advances and future directions. *Nat Rev Rheumatol*. 2017 Jun; 13(6):359-367.
3. Gravallesse EM, Schett G. Effects of the IL-23-IL-17 pathway on bone in spondyloarthritis. *Nat Rev Rheumatol*. 2018 Nov; 14(11):631-640.
4. El-Zayadi AA, Jones EA, Churchman SM, Baboolal TG, Cuthbert RJ, El-Jawhari JJ, et al. Interleukin-22 drives the proliferation, migration and osteogenic differentiation of mesenchymal stem cells: a novel cytokine that could contribute to new bone formation in spondyloarthropathies. *Rheumatology*. 2017 Mar; 56(3):488-493.
5. Tseng HW, Pitt ME, Glant TT, McRae AF, Kenna TJ, Brown MA, et al. Inflammation-driven bone formation in a mouse model of ankylosing spondylitis: sequential not parallel processes. *Arthritis Res Ther*. 2016 Jan 29; 18:35.
6. McGonagle DG, McInnes IB, Kirkham BW, Sherlock J, Moots R. The role of IL-17A in axial spondyloarthritis and psoriatic arthritis: recent advances and controversies. *Annals of the Rheumatic Diseases*. 2019 Sep; 78(9):1167-1178.
7. Sherlock JP, Joyce-Shaikh B, Turner SP, Chao CC, Sathe M, Grein J, et al. IL-23 induces spondyloarthropathy by acting on ROR-gammat+ CD3+CD4-CD8- enthesal resident T cells. *Nat Med*. 2012 Jul 1; 18(7):1069-1076.

8. van Tok MN, van Duivenvoorde LM, Kramer I, Ingold P, Pfister S, Roth L, et al. Interleukin-17A Inhibition Diminishes Inflammation and New Bone Formation in Experimental Spondyloarthritis. *Arthritis Rheumatol*. 2019 Apr; 71(4):612-625.
9. Sims AM, Timms AE, Bruges-Armas J, Burgos-Vargas R, Chou CT, Doan T, et al. Prospective meta-analysis of interleukin 1 gene complex polymorphisms confirms associations with ankylosing spondylitis. *Ann Rheum Dis*. 2008 Sep; 67(9):1305-1309.
10. van der Paardt M, Crusius JB, Garcia-Gonzalez MA, Baudoin P, Kostense PJ, Alizadeh BZ, et al. Interleukin-1beta and interleukin-1 receptor antagonist gene polymorphisms in ankylosing spondylitis. *Rheumatology (Oxford)*. 2002 Dec; 41(12):1419-1423.
11. Yamaguchi T, Olozak I, Chattopadhyay N, Butters RR, Kifor O, Scadden DT, et al. Expression of extracellular calcium (Ca²⁺)-sensing receptor in human peripheral blood monocytes. *Biochem Biophys Res Commun*. 1998 May 19; 246(2):501-506.
12. Olszak IT, Poznansky MC, Evans RH, Olson D, Kos C, Pollak MR, et al. Extracellular calcium elicits a chemokinetic response from monocytes in vitro and in vivo. *J Clin Invest*. 2000 May; 105(9):1299-1305.
13. Kanatani M, Sugimoto T, Kanzawa M, Yano S, Chihara K. High extracellular calcium inhibits osteoclast-like cell formation by directly acting on the calcium-sensing receptor existing in osteoclast precursor cells. *Biochem Biophys Res Commun*. 1999 Jul 22; 261(1):144-148.
14. Mentaverri R, Yano S, Chattopadhyay N, Petit L, Kifor O, Kamel S, et al. The calcium sensing receptor is directly involved in both osteoclast differentiation and apoptosis. *Faseb Journal*. 2006 Dec; 20(14):2562-+.
15. Goltzman D, Hendy GN. The calcium-sensing receptor in bone--mechanistic and therapeutic insights. *Nat Rev Endocrinol*. 2015 May; 11(5):298-307.
16. Shu L, Ji J, Zhu Q, Cao G, Karaplis A, Pollak MR, et al. The calcium-sensing receptor mediates bone turnover induced by dietary calcium and parathyroid hormone in neonates. *J Bone Miner Res*. 2011 May; 26(5):1057-1071.
17. Im CH, Kang EH, Ki JY, Shin DW, Choi HJ, Chang EJ, et al. Receptor activator of nuclear factor kappa B ligand-mediated osteoclastogenesis is elevated in ankylosing spondylitis. *Clin Exp Rheumatol*. 2009 Jul-Aug; 27(4):620-625.
18. Gengenbacher M, Sebald HJ, Villiger PM, Hofstetter W, Seitz M. Infliximab inhibits bone resorption by circulating osteoclast precursor cells in patients with rheumatoid arthritis and ankylosing spondylitis. *Annals of the Rheumatic Diseases*. 2008 May; 67(5):620-624.
19. Lories RJ, Derese I, Luyten FP. Inhibition of osteoclasts does not prevent joint ankylosis in a mouse model of spondyloarthritis. *Rheumatology (Oxford)*. 2008 May; 47(5):605-608.

Reviewer 3

1. This very interesting manuscript reports a potentially important mechanism of new bone formation in AS. The effects of different cytokines illustrate how this could be related to inflammatory immune mediated stimuli from several relevant cytokines. A

very important synergy in this context is that between TNF α and IL-17A. It would be very informative to test if this synergy is found in activation of the CaSR pathway.

Response:

Thank you for the careful review and kind suggestion. To validate whether TNF α and IL-17A have a synergy effect to upregulated CaSR expression, MC3T3-E1 preosteoblasts were treated with TNF α , IL-17A alone or combination of TNF α and IL-17A. The results showed that the combination of TNF α and IL-17A did not have a synergy effect to upregulated CaSR expression compared to TNF α or IL-17A alone (Figure EV4J). It indicates that each one of these cytokines is sufficient to increase the expression of CaSR and inhibition of an individual cytokine is unable to sufficiently decrease the upregulated level of CaSR. This information has been added into the revised manuscript (Page 15, Line 21-22; Page 16, Line 1-3).

Figure EV4J

(Figure EV4J) RT-qPCR analyses of CaSR expression in MC3T3-E1 cells treated with TNF α , IL-17A alone or combination of TNF α and IL-17A.

29th Sep 2020

Dear Prof. Liu,

Thank you for the submission of your revised manuscript to EMBO Molecular Medicine. We have now received the enclosed reports from the referees that were asked to re-assess it. As you will see the reviewers is now globally supportive and I am pleased to inform you that we will be able to accept your manuscript pending the following final amendments:

1) In the main manuscript file, please do the following:

- In M&M, include a statement that informed consent was obtained from all human subjects and that the experiments conformed to the principles set out in the WMA Declaration of Helsinki and the Department of Health and Human Services Belmont Report.
- In M&M, for animal work, confirm that all experiments were performed in accordance with relevant guidelines and regulations. The manuscript must include a statement in the Materials and Methods identifying the institutional and/or licensing committee approving the experiments and the licensing number when appropriate. Gender, age, origin of the animals and genetic background must be indicated, along with housing conditions.

***** Reviewer's comments *****

Referee #2 (Remarks for Author):

The authors have provided satisfactory responses to my queries.

Point-By-Point Responses to the Reviewers' comments**Referee #2**

The authors have provided satisfactory responses to my queries.

Response:

Thank you for the careful review and recognition to our works.

Point-By-Point Responses to the Editors' comments

- In M&M, include a statement that informed consent was obtained from all human subjects and that the experiments conformed to the principles set out in the WMA Declaration of Helsinki and the Department of Health and Human Services Belmont Report.
- In M&M, for animal work, confirm that all experiments were performed in accordance with relevant guidelines and regulations. The manuscript must include a statement in the Materials and Methods identifying the institutional and/or licensing committee approving the experiments and the licensing number when appropriate. Gender, age, origin of the animals and genetic background must be indicated, along with housing conditions.
- A statement that informed consent was obtained from all human subjects and that the experiments conformed to the principles set out in the WMA Declaration of Helsinki and the Department of Health and Human Services Belmont Report have been added in 'M&M' section. (Page 25, Line 1-3)
- We confirmed that all experiments were performed in accordance with relevant guidelines and regulations. Gender, age, origin of the animals, genetic background and housing conditions was indicated in 'M&M' section. In addition, a statement that The Medical Ethics Committee of the First Affiliated Hospital of Sun Yat-sen University approved the procedures performed in this study was added in 'M&M' section. (Page 25, Line 5-8; Page 26, Line 22; Page 27, Line1)

The authors performed the requested changes.

Corresponding Author Name: Hui Liu

Manuscript Number: EMM-2020-12109-V2